# Low-pass Personalized Subgraph Federated Recommendation

**Wooseok Sim, Hogun Park**[*]
Department of Artificial Intelligence
Sungkyunkwan University
Suwon, Republic of Korea
{dntjr41, hogunpark}@skku.edu

## ABSTRACT

Federated Recommender Systems (FRS) preserve privacy by training decentralized models on client-specific user-item subgraphs without sharing raw data. However, FRS faces a unique challenge: *subgraph structural imbalance*, where drastic variations in subgraph scale (user/item counts) and connectivity (item degree) misalign client representations, making it challenging to train a robust model that respects each client's unique structural characteristics. To address this, we propose a **L**ow-pass **P**ersonalized **S**ubgraph **Fed**erated recommender system (**LPSFed**). LPSFed leverages graph Fourier transforms and low-pass spectral filtering to extract low-frequency structural signals that remain stable across subgraphs of varying size and degree, allowing robust personalized parameter updates guided by similarity to a neutral structural anchor. Additionally, we leverage a localized popularity bias-aware margin that captures item-degree imbalance within each subgraph and incorporates it into a personalized bias correction term to mitigate recommendation bias. Supported by theoretical analysis and validated on five real-world datasets, LPSFed achieves superior recommendation accuracy and enhances model robustness.

## 1 INTRODUCTION

Federated Recommender Systems (FRS) play a crucial role in preserving privacy while maintaining recommendation quality. For example, large e-commerce platforms (e.g., Amazon, eBay) can treat user-item interactions via subgraphs, where each client corresponds to a specific country or region and accesses only localized data. Under a Federated Learning (FL) framework, models exchange parameters instead of raw user data, significantly enhancing data privacy (McMahan et al., 2017). Existing FRS approaches include matrix factorization-based methods, e.g., (Chai et al., 2020), personalized models (Liu et al., 2022; Zhang et al., 2023a; Li et al., 2024), and graph-based methods, e.g., FedPerGNN (Wu et al., 2022; Qu et al., 2023) that construct subgraphs at the server; however, these methods largely assume comparable client subgraph structures. Meanwhile, decentralized learning naturally introduces substantial heterogeneity across clients from variations in the size and data distributions within local datasets. Heterogeneity has been widely studied in other FL tasks. In image classification FL, it appears through differences in local image quantities and skewed class distributions (Duan et al., 2020; Hsu et al., 2020; Wang et al., 2020a). In graph-based node classification FL, heterogeneity arises as client subgraphs differ in class distributions and class-driven graph topology, with some classes forming dense clusters and others appearing isolated or sparse (Fu et al., 2024; Kong et al., 2024; Li et al., 2023; Tan et al., 2025b).

In subgraph-based FRS, the key challenge is *subgraph structural imbalance*, significant variations in client subgraph size (user/item counts) and connectivity (item degree). This divergence is problematic for spatial Graph Neural Networks (GNNs), such as PinSage, NGCF, and LightGCN (Ying et al., 2018; Wang et al., 2019; He et al., 2020), because their multi-hop message passing is highly sensitive to local topology. Consequently, when client subgraphs have vastly different structures, their locally trained models produce misaligned representations, destabilizing federated updates and degrading

---

[*]Corresponding author.

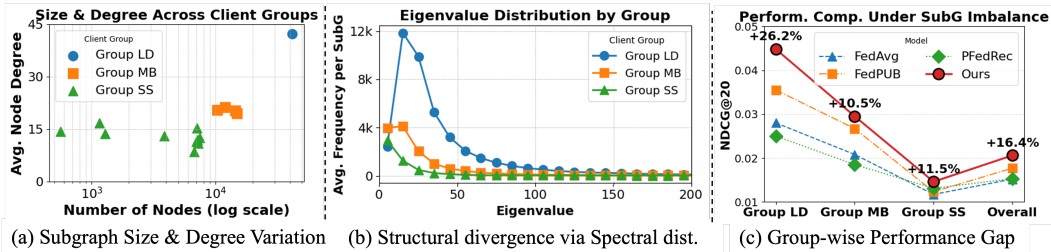

**Figure 1:** Empirical observations from the federated recommender systems on the *Amazon-Book* (He et al., 2020) dataset. (a) Subgraph size-degree variation: each point is one of 15 client subgraphs, partitioned using spectral clustering, grouped into **L**arge-**D**ense (**LD**), **M**edium-**B**alanced (**MB**), **S**mall-**S**parse (**SS**) by node count and average degree. (b) Structural divergence: Laplacian eigenvalue histograms averaged over each group, highlighting distinct spectral signals. (c) Group-wise Performance Gap: NDCG@20 for FedAvg (McMahan et al., 2017), PFedRec (Zhang et al., 2023a), FedPUB (Baek et al., 2023), and Ours across groups, showing that performance varies significantly depending on subgraph structure. Detailed experimental results in Table 2.

recommendation quality. Most existing GNN-based FRS methods adopt such spatial architectures without explicitly accounting for structural imbalance across decentralized subgraphs. Spectral FL methods (Tan et al., 2024; Yu, 2025; Tan et al., 2025a) have been explored as an alternative to mitigate client heterogeneity by aligning low-frequency structural components. However, their direct application to FRS is challenging since recommendation graphs are bipartite, lack informative node features, and exhibit severe degree skewness, unlike the homogeneous graph structures typically assumed in spectral FL studies on citation or social networks.

Beyond representation misalignment, *subgraph structural imbalance* also amplifies *localized popularity bias* in FRS. Specifically, in dense clients with high average item degree, GNN aggregations are dominated by high-degree hub items, overshadowing the long-tail items. Conversely, in sparse clients with low average item degree, the scarcity of connections forces the model to rely on a narrow set of relatively higher-degree items, leading to unstable training and poor long-tail learning (Abdollahpouri et al., 2019; Mansoury et al., 2020; Gao et al., 2022; Zhang et al., 2024; Lin et al., 2025). As training progresses, this bias fosters a self-reinforcing feedback loop, further consolidating the dominance of popular items (Chaney et al., 2018; Klimashevskaia et al., 2024). Since client isolation in federated settings prevents the sharing of global context that could mitigate this effect, it is crucial to build adaptive strategies tailored to each subgraph's structural characteristics.

Figure 1 highlights three key aspects we examine on *Amazon-Book* (He et al., 2020) dataset in an FL setting. To simulate this scenario with controlled structural diversity, we partition the global interaction graph into 15 clients using spectral clustering, resulting in structurally distinct groups. The figure illustrates that variations in client subgraph size and degree lead to pronounced spectral divergence, which in turn corresponds to a consistent decrease in NDCG@20 for existing baseline methods. These observations demonstrate that *subgraph structural imbalance* directly degrades FRS performance. By leveraging spectral information like Laplacian eigenvalue distributions, our method consistently improves performance across all client groups, demonstrating the effectiveness of spectral signals under structural heterogeneity.

Motivated by these observations, we propose **Low-pass Personalized Subgraph Federated Recommendation** (**LPSFed**), a robust personalized FRS framework that addresses structural imbalance through two synergistic components. **(1)** We apply low-pass spectral filtering to each client subgraph to extract its dominant low-frequency structural signal, which reflects the core connectivity pattern with minimal sensitivity to scale and noise. These signals are used to compute personalized structural similarities between each client and the global model, guiding adaptive parameter updates referenced against a neutral structural anchor across heterogeneous subgraphs. **(2)** We incorporate a localized popularity bias-aware margin that captures variations in item-degree distributions across subgraphs and applies a personalized correction term during local update.

We evaluate LPSFed on five real-world datasets across diverse FRS scenarios and compare it against representative baselines. The results demonstrate that our method consistently achieves more stable and accurate recommendations. These findings highlight the importance of addressing structural imbalance for improving personalized federated recommendations in subgraph-based settings.

## 2 PRELIMINARY

### 2.1 FEDERATED RECOMMENDER SYSTEM

In a Federated Recommender System (FRS), user-item interactions are decentralized into $C$ distinct subgraphs $\{G_1, ..., G_C\}$. For a given client $c$, we denote its local subgraph as $G_c = (\mathcal{V}_c, \mathcal{E}_c)$, which consists of the node set $\mathcal{V}_c = \mathcal{U} \cup \mathcal{I}$ and the edge set $\mathcal{E}_c$. Here, $\mathcal{U} = \{u_1, ..., u_M\}$ and $\mathcal{I} = \{i_1, ..., i_N\}$ represent the sets of $M$ users and $N$ items within that client $c$. The recommendation task is to predict a preference score $\hat{y}_{ui}$ for unobserved user-item pairs $(u, i)$ and generate a top-$K$ ranked list.

### 2.2 GRAPH FOURIER TRANSFORM (GFT)

The Graph Fourier Transform (GFT) (Ramakrishna et al., 2020; Isufi et al., 2024) extends classical Fourier analysis to graph-structured data by leveraging the graph Laplacian matrix $\mathbf{L} = \mathbf{I} - \mathbf{D}^{-\frac{1}{2}} \mathbf{A} \mathbf{D}^{-\frac{1}{2}}$, where the adjacency matrix $\mathbf{A} = \begin{bmatrix} 0 & \mathbf{R} \\ \mathbf{R}^\mathsf{T} & 0 \end{bmatrix}$ with $\mathbf{R} \in \mathbb{R}^{M \times N}$. $\mathbf{D}$ is the diagonal degree matrix with $\mathbf{D}_{ii} = \sum_j \mathbf{A}_{ij}$, and $\mathbf{I}$ is the identity matrix. We define the node embeddings $\mathbf{Z} \in \mathbb{R}^{(M+N) \times D}$ by stacking user $\mathbf{U}^{M \times D}$ and item $\mathbf{V}^{N \times D}$ embeddings, where $D$ represents the embedding dimensionality. By eigendecomposition $\mathbf{L} = \mathbf{P} \mathbf{\Lambda} \mathbf{P}^\mathsf{T}$, where $\mathbf{P}$ is the matrix of eigenvectors (frequency bases) and $\mathbf{\Lambda} = diag([\lambda_1, \lambda_2, \ldots, \lambda_{M+N}])$ is the diagonal matrix of eigenvalues (frequencies). The GFT of an embedding matrix $\mathbf{Z}$ is $\tilde{\mathbf{Z}} = \mathcal{F}_g(\mathbf{Z}) = \mathbf{P}^\mathsf{T} \mathbf{Z}$, while the inverse GFT is: $\mathbf{Z} = \mathcal{F}_g^{-1}(\tilde{\mathbf{Z}}) = \mathbf{P}\tilde{\mathbf{Z}}$. Through these transforms, graph signals are decomposed into frequency components aligned with the graph's topology, enabling selective reconstruction or modification for filtering and structural analysis tasks.

### 2.3 LOW-PASS GRAPH FILTER & CONVOLUTION

Low-pass graph filters (Nt & Maehara, 2019; Yu & Qin, 2020; Liu et al., 2023) preserve meaningful low-frequency structures by suppressing high-frequency noise. The filter is defined by a simple gate function, $\tilde{\mathbf{f}} = \begin{bmatrix} 1^{\Phi} \\ 0^{M+N-\Phi} \end{bmatrix}$, where $\Phi$ is the cut-off frequency. The Low-pass Collaborative Filter (LCF) (Yu et al., 2022) is applied as: $LCF(\mathbf{Z}) = \mathcal{F}_g^{-1}(diag(\tilde{\mathbf{f}}) \cdot \mathcal{F}_g(\mathbf{Z})) = \bar{\mathbf{P}}\bar{\mathbf{P}}^\mathsf{T}\mathbf{Z}$, where $\bar{\mathbf{P}} = \mathbf{P}_{*,1:\Phi}$ contains the first $\Phi$ eigenvectors. When $\Phi = M + N$, the filter becomes all-pass, retaining all frequencies. Adjusting $\Phi$, LCF selectively preserves low-frequency signals while minimizing the impact of high-frequency noise.

Low-pass Graph Convolutional Network (LGCN (Yu et al., 2022)) utilizes these graph filters for efficient convolution operations, leveraging the convolution theorem (Barrett & Wilde, 1960). Given an embedding matrix $\mathbf{Z}$ and convolution kernel $\mathbf{k} \in \mathbb{R}^{M+N}$, graph convolution is defined as:

$$\mathbf{Z} *_g \mathbf{k} = \mathcal{F}_g^{-1}(diag(\tilde{\mathbf{k}}) \cdot \mathcal{F}_g(\mathbf{Z})) = \mathbf{P} diag(\tilde{\mathbf{k}}) \mathbf{P}^\mathsf{T} \mathbf{Z}, \tag{1}$$

where $*_g$ represents graph convolution, and $\tilde{\mathbf{k}}$ is the kernel in the frequency domain. Combining graph convolution with low-pass filtering results in a low-pass convolution:

$$\mathbf{Z} \bar{*}_g \mathbf{k} = \mathcal{F}_g^{-1}(diag(\tilde{\mathbf{f}}) \cdot diag(\tilde{\mathbf{k}}) \cdot \mathcal{F}_g(\mathbf{Z})) = \bar{\mathbf{P}} diag(\bar{\mathbf{k}}) \bar{\mathbf{P}}^\mathsf{T} \mathbf{Z}, \tag{2}$$

where $\bar{*}_g$ denotes low-pass graph convolution, and $\bar{\mathbf{k}} = \tilde{\mathbf{k}}_{1:\Phi}$ represents the truncated convolution kernel, ensuring computational efficiency by using only the first $\Phi$ eigenvectors, compared to standard graph convolution. Its time complexity is $\mathcal{O}(n\Phi^2)$, where $n$ denotes the number of non-zero elements in $\mathbf{L}$. In practice, $\Phi \ll M + N$ and $n \ll (M + N)^2$, making the computation efficient for sparse graphs. Instead of performing a full eigendecomposition, we compute only the first $\Phi$ eigenvectors using a Lanczos solver (Grimes et al., 1994), which leverages graph sparsity. This computation is performed once during the preprocessing stage, prior to training, and does not affect per-round update or communication costs. LGCN starts with an initial embedding layer $\mathbf{Z}^{(0)}$, followed by $L$ graph convolution layers. Each $l$-th layer updates feature maps as:

$$\mathbf{Z}^{(l)} = \bar{\mathbf{P}} diag(\bar{\mathbf{k}}^{(l)}) \bar{\mathbf{P}}^\mathsf{T} \mathbf{Z}^{(l-1)}. \tag{3}$$

After $L$ iterations, embeddings are pooled across all layers to produce the final predictive embeddings.

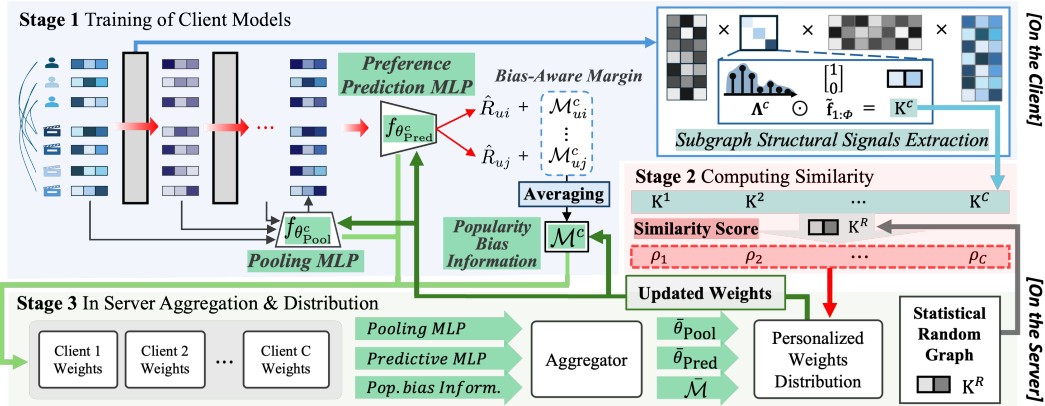

**Figure 2:** Overview of LPSFed - On the Client: Stage 1 applies low-pass GCN and a localized popularity bias-aware loss to train client subgraphs. Stage 2 computes similarities between each client subgraph and a server-provided random graph using structural signals. On the Server: Stage 3 aggregates client parameters and distributes them based on personalized similarity scores. Colored arrows indicate stage-wise interactions.

# 3 METHODOLOGY

In this section, we propose the ***Low-pass Personalized Subgraph Federated Recommendation*** (**LPSFed**) which encompasses three key components outlined in Figure 2 and Algorithm 1:

- **[In Client] Stage 1: Training of Client Models** - Applies low-pass GCN and bias-aware loss to train subgraph models and extract structural signals, and localized popularity bias information.
- **[In Client] Stage 2: Computing Structural Similarity** - Computes the structural similarity by comparing the client's structural signals against the server-provided neutral structural anchor.
- **[In Server] Stage 3: Aggregating and Distributing Parameters on the Server** - Aggregates client parameters on the server and distributes them based on personalized similarity scores.

## 3.1 TRAINING OF CLIENT MODELS

**Client Models.** (Stage 1 of Figure 2) Each client independently processes its local subgraph using a multi-layer Low-pass Graph Convolutional Network (LGCN) (Yu et al., 2022). Within this LGCN, we insert two Multi-Layer Perceptron (MLP) modules: **Pooling MLP** ($f_{\theta_{\text{Pool}}^c}$) merges per-layer user and item node embeddings into a client-specific single representation per node:

$$\mathbf{Z}^c = f_{\theta_{\text{Pool}}^c}(\{\mathbf{Z}^{(l)}\}_{l=0}^L), \quad \text{for } c = 1, \ldots, C, \tag{4}$$

where $C$ is the number of clients, and $\mathbf{Z}^{(l)}$ (Eq. 3) represents the node embedding matrix at layer $l$. $\mathbf{U}_u^c$ and $\mathbf{V}_i^c$ are the pooled user/item embeddings for client $c$, which is split into user embeddings $\mathbf{U}^c = \mathbf{Z}_{1:M}^c$ and item embeddings $\mathbf{V}^c = \mathbf{Z}_{M+1:M+N}^c$.

The **Predictive MLP** ($f_{\theta_{\text{Pred}}^c}$) computes a preference score from the user-item pair's pooled embeddings, which is then converted into the final preference angle $\hat{R}_{ui}$:

$$\hat{R}_{ui} = \arccos(\tanh(f_{\theta_{\text{Pred}}^c}([\mathbf{U}_u^c, \mathbf{V}_i^c, \mathbf{U}_u^c \odot \mathbf{V}_i^c]))), \quad \text{for } c = 1, \ldots, C, \tag{5}$$

where $f_{\theta_{\text{Pred}}^c}$ is a client-specific MLP that outputs an unbounded scalar preference score, which is then converted into the final preference angle $\hat{R}_{ui}$ bounded in the range $[0, \pi]$.

**Subgraph Structural Signals Extraction.** We extract each client's *denoised structural signals* by constructing a low-pass convolution kernel distribution $\bar{\mathbf{k}}$ from its subgraph's eigenvalue spectrum:

$$\mathbf{K}^c = \bar{\mathbf{k}}^c = \tilde{\mathbf{k}}_{1:\Phi}^c = \mathbf{\Lambda}^c \odot \tilde{\mathbf{f}}_{1:\Phi}, \quad \text{for } c = 1, \ldots, C, \tag{6}$$

where $\mathbf{\Lambda}^c$ is the diagonal eigenvalue matrix of client $c$, and $\tilde{\mathbf{f}}_{1:\Phi}$ is a spectral filter that retains only the first $\Phi$ components. This low-pass kernel effectively suppresses high-frequency noise while

preserving the core structural patterns of each client's subgraph, which are critical for capturing meaningful connectivity and mitigating the impact of structural heterogeneity across clients (Shuman et al., 2013). By integrating $\mathbf{K}^c$ into federated learning, our model accurately extracts and exploits the *denoised structural signals* of each client's subgraph.

**Localized Popularity Bias-aware Optimization.** To address *popularity bias*, our framework utilizes the bias information in two ways: (1) as an auxiliary contrastive loss to regularize the bias embedding space, and (2) as an adaptive margin in the main recommendation loss (Zhang et al., 2022). First, user and item popularity scores $(p_u, p_i)$ are encoded into $d$-dimensional embeddings via $f_{\psi_{bias}}(p_u)$ and $f_{\phi_{bias}}(p_i)$. The bias score $s(\cdot)$ is the cosine similarity between these embeddings, defined as $s(f_{\psi_{bias}}(p_u), f_{\phi_{bias}}(p_i)) = \cos(\hat{\xi}_{ui})$, where the scalar $\hat{\xi}_{ui}$ is the angle between the two embedding vectors. This score is first used in an *auxiliary bias contrastive loss*, $\mathcal{L}_{bias}$, designed to train the bias encoders $(f_{\psi_{bias}}, f_{\phi_{bias}})$:

$$\mathcal{L}_{bias} = - \sum_{(u,i) \in O^+} \log \frac{\exp(\cos(\hat{\xi}_{ui})/\tau)}{\exp(\cos(\hat{\xi}_{ui})/\tau) + \sum_{j \in N_u} \exp(\cos(\hat{\xi}_{uj})/\tau)}, \tag{7}$$

where $\tau$ is the temperature and $N_u$ is the negative set for user $u$. Second, we use the *same bias angle* $\hat{\xi}_{ui}$ to construct an adaptive margin for the main recommendation task:

$$\mathcal{M}_{ui}^c = \min\{\gamma \cdot \hat{\xi}_{ui}, \pi - \hat{R}_{ui}\}, \quad \text{for } c = 1, \ldots, C, \tag{8}$$

where $\gamma$ controls the margin strength and $\pi - \hat{R}_{ui}$ enforces a monotonic decrease. This locally-computed margin is then refined by interpolating it with a personalized global context (detailed in (Eq. 15), creating the refined margin $\widetilde{\mathcal{M}}_{ui}^c$ that is used in the Bias-aware Contrastive (BC)-loss:

$$\mathcal{L}_{BC} = - \sum_{(u,i) \in O^+} \log \frac{\exp(\cos(\hat{R}_{ui} + \widetilde{\mathcal{M}}_{ui}^c)/\tau)}{\exp(\cos(\hat{R}_{ui} + \widetilde{\mathcal{M}}_{ui}^c)/\tau) + \sum_{j \in N_u} \exp(\cos(\hat{R}_{uj})/\tau)}. \tag{9}$$

This formulation adaptively penalizes over-recommended items and encourages long-tail exposure, mitigating localized popularity bias.

**Localized Popularity Bias Information Computation.** Each client aggregates its popularity bias-aware margins into a *single representative value* to preserve privacy and ensure effective global utilization. The average margin for client $c$ is computed as:

$$\mathcal{M}^c = \frac{1}{MN} \sum_{u=1}^{M} \sum_{i=1}^{N} \mathcal{M}_{ui}^c, \quad \text{for } c = 1, \ldots, C, \tag{10}$$

where $C$, $M$, and $N$ are the number of clients, users, and items, respectively. This scalar $\mathcal{M}^c$ represents the overall *localized popularity bias information* within each client's subgraph and is deliberately aggregated into a single non-invertible value to prevent the exposure of user- or item-level bias characteristics, thereby minimizing privacy risks. The server then utilizes $\mathcal{M}^c$ to construct the *global bias context*, which guides inter-client bias mitigation. In contrast, sharing the bias encoder parameters $(f_{\psi_{bias}}, f_{\phi_{bias}})$ is avoided because they encode highly fine-grained, client-specific preference patterns. Exchanging such detailed parameters could inadvertently expose sensitive interaction information and lead to instability when aggregated across heterogeneous clients. Additional details on the procedure TRAINCLIENTMODEL in Algorithm 3, line 1.

## 3.2 COMPUTING STRUCTURAL SIMILARITY

(Stage 2 of Figure 2) Since the server cannot access raw subgraph data, it aggregates high-level client statistics (e.g., average node/edge counts) during initialization. Using these statistics, it generates an informed reference graph $G_R$ via Erdos-Renyi (ER) or GNMK models (Erdős et al., 1960; Knuth, 1977) at each global epoch, providing a *neutral structural anchor* for similarity comparison. Under strict privacy constraints, this step may be omitted, allowing the server to build $G_R$ without relying on client-specific statistics. The server performs a GCN on $G_R$ and generates a convolution kernel

distribution $\mathbf{K}^R$ which is distributed to each client. Each client, without sharing its local kernel $\mathbf{K}^c$, computes its **structural similarity** by using the KL-divergence (Hershey & Olsen, 2007) between its local kernel $\mathbf{K}^c$ (Eq. 6) and the reference kernel $\mathbf{K}^R$:

$$\rho_c = D_{KL}(\mathbf{K}^R \parallel \mathbf{K}^c) = \sum_{i=1}^{\Phi} \mathbf{K}^R(i) \log\left(\frac{\mathbf{K}^R(i)}{\mathbf{K}^c(i)}\right), \quad \text{for } c = 1, \ldots, C. \tag{11}$$

To align these scores across clients, the server normalizes the similarities via min-max normalization:

$$\bar{\rho}_c = 1 - \frac{(\rho_c - \min(\rho))}{(\max(\rho) - \min(\rho))}, \quad \text{where } \rho = \{\rho_1, \ldots, \rho_C\}. \tag{12}$$

The normalized similarity $\bar{\rho}_c$ quantifies each client's structural alignment to the reference graph. This score guides the personalized FL updates and mitigates *subgraph structural imbalance* by reducing the impact of highly divergent clients that would otherwise introduce misaligned representations into the global model. Further details on the procedure COMPUTESIM. in Algorithm 3.

## 3.3 AGGREGATING AND DISTRIBUTING PARAMETERS ON THE SERVER

**Initialization.** In the initial communication round, the server initializes the learning environment by distributing global parameters $\bar{\theta}$ to all clients, ensuring a uniform starting point $\theta^c \leftarrow \bar{\theta}$ for all clients $c$. Refer to Algorithm 1, line 1.

**Aggregation.** (Stage 3 of Figure 2) The server aggregates two model parameters and scalar bias signals: pooling MLP's $\theta_{\text{Pool}}^c$ (Eq. 4), predictive MLP's $\theta_{\text{Pred}}^c$ (Eq. 5), and averaged margin $\mathcal{M}^c$ (Eq. 10). Aggregation involves computing the mean across all clients:

$$\bar{\theta}_{\text{Pool}} = \frac{1}{C}\sum_{c=1}^{C}\theta_{\text{Pool}}^c, \quad \bar{\theta}_{\text{Pred}} = \frac{1}{C}\sum_{c=1}^{C}\theta_{\text{Pred}}^c, \quad \bar{\mathcal{M}} = \frac{1}{C}\sum_{c=1}^{C}\mathcal{M}^c. \tag{13}$$

where $C$ is the number of clients. These procedures are run every global training epoch, as described in **Global Training Loop** part in Algorithm 2, line 5.

**Distribution.** The server distributes updated model parameters to each client by adjusting them according to the client's normalized similarity score $\bar{\rho}_c$ (Eq. 12). This score balances the influence of the global model and the client's local model during update:

$$\begin{aligned}
\theta_{\text{Pool},\{\text{updated}\}}^c &= (\bar{\theta}_{\text{Pool}} \times \bar{\rho}_c) + (\theta_{\text{Pool}}^c \times (1 - \bar{\rho}_c)), \\
\theta_{\text{Pred},\{\text{updated}\}}^c &= (\bar{\theta}_{\text{Pred}} \times \bar{\rho}_c) + (\theta_{\text{Pred}}^c \times (1 - \bar{\rho}_c)), \\
\mathcal{M}_{\{\text{updated}\}}^c &= (\bar{\mathcal{M}} \times \bar{\rho}_c) + (\mathcal{M}^c \times (1 - \bar{\rho}_c)).
\end{aligned} \tag{14}$$

While the updated parameters $\theta_{\{\text{updated}\}}^c$ are used directly, the distributed margin $\mathcal{M}_{\{\text{updated}\}}^c$ provides global bias context to the client's next local training round. The client creates a refined margin $\widetilde{\mathcal{M}}_{ui}^c$ by interpolating this received global value with its newly-computed local margin $\mathcal{M}_{ui}^c$ (Eq. 8):

$$\widetilde{\mathcal{M}}_{ui}^c = \omega\mathcal{M}_{\{\text{updated}\}}^c + (1 - \omega)\mathcal{M}_{ui}^c, \tag{15}$$

where $\omega \in [0, 1]$ balances global guidance and local specificity. It is this *refined margin* that is then fed into the BC-loss (Eq. 9), ensuring the final loss reflects both global knowledge and structural uniqueness. For implementation details, see the procedure UPDATECLIENT in Algorithm 2, line 11.

## 3.4 THEORETICAL ANALYSIS

This section presents a theoretical analysis of our approach to *subgraph structural imbalance*. We provide a dual justification: first, we prove that our similarity metric (based on filtered spectrum) is a stable measure of structural similarity, and second, we analyze how our low-pass filtering method serves as a powerful spectral regularizer. We demonstrate that both our metric and method are fundamentally governed by the underlying graph structure. Detailed proofs in Appendix B.

---

**Algorithm 1** LPSFed: Low-pass Personalized Subgraph Federated Recommendation

**Notations.**
Client $c \in \{1, ..., C\}$, Server $S$, local/global epochs $e_c, e_g$, parameters $\theta_{\text{All}} = \bar{\theta}_{\text{Pool}}, \bar{\theta}_{\text{Pred}}, \bar{\mathcal{M}}$

1: **Server Initialization:**
2: 1) Initialize global model $\bar{\theta}_{\text{All}}$         2) Distribute parameters to clients: $\theta_{\text{All}}^c \leftarrow \bar{\theta}_{\text{All}}$
3: 3) Aggregate statistics from all clients

---

**Algorithm 2** Global Training Loop

1: **for** $epoch = 1$ to $e_g$ **do**      #[Stage 3]
2:      Generate a statistical random graph $G_R$
3:      Train global model with $G_R \Rightarrow \mathbf{K}^R$
4:      **for all** client $c$ (in parallel) **do**
5:          $\theta_{\text{All}}^c, \mathbf{K}^c \leftarrow \text{TRAINCLIENTMODEL}(c)$
6:          $\rho_c \leftarrow \text{COMPUTESIM.}(c, \mathbf{K}^R, \mathbf{K}^c)$
7:      $\bar{\theta}_{\text{All}} \leftarrow \frac{1}{C} \sum_c \theta_{\text{All}}^c$
8:      Normalize: $\bar{\rho}_c \leftarrow 1 - \frac{\rho_c - \min(\rho)}{\max(\rho) - \min(\rho)}$
9:      **for all** client $c$ **do**
10:        $\theta_{\text{new}}^c \leftarrow \bar{\theta}_{\text{All}} \cdot \bar{\rho}_c + \theta_{\text{All}}^c \cdot (1 - \bar{\rho}_c)$
11:        $\text{UPDATECLIENT}(c, \theta_{\text{new}}^c)$

**Algorithm 3** Procedures

1: **Proc.** $\text{TRAINCLIENTMODEL}(c)$    #[Stage 1]
2:      **for** $epoch = 1$ to $e_c$ **do**
3:          Train $\theta_{\text{All}}^c$, kernel $\mathbf{K}^c$ with GCN
4:      **return** $\theta_{\text{All}}^c, \mathbf{K}^c$
5:
6: **Proc.** $\text{COMPUTESIM.}(c, \mathbf{K}^R, \mathbf{K}^c)$    #[Stage 2]
7:      Compute $KL$-divergence $\mathbf{K}^R$ & $\mathbf{K}^c$
8:      **return** $\rho_c$
9:
10: **Proc.** $\text{UPDATECLIENT}(c, \theta_{\text{new}}^c)$
11:      Apply updated parameters to client $c$

---

**Theorem 3.1** (Structural Comparison via Spectral Distributions). *Let $G_1 = (\mathcal{V}_1, \mathcal{E}_1)$ and $G_2 = (\mathcal{V}_2, \mathcal{E}_2)$ be graphs with $n_1 = |\mathcal{V}_1|$ and $n_2 = |\mathcal{V}_2|$ nodes and $k$ communities each. $\mathcal{E}_1$ and $\mathcal{E}_2$ are sets of edges of $G_1$ and $G_2$, respectively. Moreover, $\Phi$ $(< n)$ denotes the number of eigenvalues below the cut-off frequency $\lambda$. Let $\mathbf{K}^1$ and $\mathbf{K}^2$ be their respective low-pass filtered eigenvalue distributions:*

$$\mathbf{K}^j(i) = \frac{\lambda_i^{(j)}}{\sum_{q=1}^{\Phi} \lambda_q^{(j)}}, \quad q \leq \Phi, j \in \{1, 2\}. \tag{16}$$

*Under Assumption 1, let $D_{struct} = D_{KL}(\mathbf{K}_{struct}^1 \| \mathbf{K}_{struct}^2)$ denote the KL-divergence between the idealized distributions $\mathbf{K}_{struct}^1$ and $\mathbf{K}_{struct}^2$ corresponding to the $k$-community structures of $G_1$ and $G_2$, respectively. $c, \epsilon > 0$ are constants. Then, (1) The KL-divergence converges to a limiting value $D^*$ with probability:*

$$\mathbb{P}(|D_{KL}(\mathbf{K}^1 \| \mathbf{K}^2) - D^*| > \epsilon) \leq 4\Phi \exp(-c \min(n_1, n_2)\epsilon^2/8). \tag{17}$$

*(2) The structural similarity is preserved with error:*

$$|D_{KL}(\mathbf{K}^1 \| \mathbf{K}^2) - D_{struct}| \leq \frac{C}{\min(\delta_1, \delta_2)}, \tag{18}$$

*where $C$ is a constant and $\delta_j$ is the eigengap $\lambda_{k+1}^{(j)} - \lambda_k^{(j)}$ for graph $G_j$.*

The theorem above shows that low-pass spectral comparison stabilizes cross-graph similarity. The KL-divergence between filtered eigenvalue distributions converges exponentially, with accuracy bounded by the smallest eigengap, indicating that clearer community separation yields more precise similarity. Therefore, the filtered spectral distribution provides a theoretically grounded basis for using $D_{KL}(\mathbf{K}^R \| \mathbf{K}^c)$ as a principled proxy for structural divergence.

**Theorem 3.2** (Spectral Regularization). *Let $G = (\mathcal{V}, \mathcal{E})$ be a client graph with $n = |\mathcal{V}|$ and the eigengap $\delta = \lambda_{k+1} - \lambda_k > 0$, where $k$ is the number of communities in the graph $G$. Denote the low-pass filtered node embedding and the $k$-community subspace embeddings by $\mathbf{z}, \mathbf{z}^* \in \mathbb{R}^n$, respectively. Assume the raw user/item embeddings are bounded by $\|\mathbf{U}_u\|_2, \|\mathbf{V}_i\|_2 \leq r$, where $r > 0$ is the embedding $\ell_2$ norm bound.*

$$|\operatorname{Var}(\mathbf{z}) - \operatorname{Var}(\mathbf{z}^*)| \leq \frac{C'}{\delta}, \tag{19}$$

*where $\operatorname{Var}(\cdot)$ denotes graph smoothness and $C' = 32\sqrt{k}r^2$.*

Our low-pass filtering method serves as a personalized spectral regularizer that bounds representation variance by the eigengap $\delta$, while our similarity metric $\rho_c$ is also a stable measure governed by this same underlying graph property. This shared governance by the eigengap provides the theoretical justification for using $\bar{\rho}_c$ (Eq. 12) as our core personalization weight.

**Table 1:** The overall performance comparison Recall@20 and NDCG@20 on five datasets, with the best scores highlighted in **bold** and the second-best in underlined. The *improvement* row highlights the gains achieved by our model (**LPSFed**) compared to the second-best-performing model.

| Dataset | Amazon-Book | | Gowalla | | Movielens-1M | | Yelp2018 | | Tmall-Buy | |
|---|---|---|---|---|---|---|---|---|---|---|
| Model | Recall | NDCG | Recall | NDCG | Recall | NDCG | Recall | NDCG | Recall | NDCG |
| FedAvg | 0.0642 | 0.0312 | 0.1425 | 0.0660 | 0.2454 | 0.1240 | 0.0721 | 0.0292 | 0.0317 | 0.0164 |
| FedPUB | 0.0633 | 0.0322 | 0.1433 | 0.0667 | 0.2558 | 0.1209 | 0.0684 | 0.0296 | 0.0333 | 0.0180 |
| FedMF | 0.0153 | 0.0072 | 0.0765 | 0.0409 | 0.1679 | 0.0906 | 0.0318 | 0.0150 | 0.0122 | 0.0063 |
| F2MF | 0.0451 | 0.0225 | 0.0961 | 0.0565 | 0.1788 | 0.1120 | 0.0510 | 0.0247 | 0.0207 | 0.0132 |
| PFedRec | 0.0713 | 0.0242 | 0.1371 | 0.0478 | 0.1508 | 0.0997 | 0.0750 | 0.0225 | 0.0323 | 0.0170 |
| FedRAP | 0.0082 | 0.0090 | 0.0340 | 0.0458 | 0.0550 | 0.0389 | 0.0129 | 0.0249 | 0.0046 | 0.0052 |
| FedPerGNN | 0.0035 | 0.0026 | 0.0958 | 0.0777 | 0.1332 | 0.1124 | 0.0252 | 0.0220 | 0.0032 | 0.0019 |
| FedHGNN | 0.0647 | 0.0298 | 0.1230 | 0.0608 | 0.2163 | 0.1031 | 0.0721 | 0.0268 | 0.0362 | 0.0171 |
| FedSSP | 0.0649 | 0.0356 | 0.1528 | 0.0729 | 0.2564 | 0.1265 | 0.0733 | 0.0315 | 0.0370 | 0.0186 |
| LPSFed (BPR) | 0.0643 | 0.0322 | 0.1529 | 0.0711 | 0.2604 | 0.1281 | 0.0769 | 0.0301 | 0.0362 | 0.0186 |
| **LPSFed** | **0.0738** | **0.0442** | **0.1621** | **0.0909** | **0.2646** | **0.1342** | **0.0783** | **0.0379** | **0.0385** | **0.0218** |
| Improvement | ↑3.5% | ↑24.2% | ↑6.1% | ↑17% | ↑3.2% | ↑6.1% | ↑4.4% | ↑20.3% | ↑4.1% | ↑17.2% |

# 4 EXPERIMENTS

## 4.1 EXPERIMENTAL SETTINGS

**Datasets.** We evaluated our model on five real-world datasets: *Amazon-Book*, *Gowalla* (He et al., 2020), *Movielens-1M* (mov), *Yelp2018* (yel), and *Tmall-Buy* (Tma). Each dataset was split into training, validation, and test sets in an 8:1:1 ratio.

**Baselines.** Nine baselines such as FedAvg, FedPUB, FedMF, F2MF, PFedRec, FedRAP, Fed-PerGNN, FedHGNN, and FedSSP (McMahan et al., 2017; Baek et al., 2023; Chai et al., 2020; Liu et al., 2022; Zhang et al., 2023a; Li et al., 2024; Wu et al., 2022; Yan et al., 2024; Tan et al., 2024) are used for comparisons. These include standard FL, federated matrix factorization, personalized FRS, fairness-aware FRS, and Spectral-based FL methods. We used *Recall@20* and *NDCG@20* to measure recommendation accuracy, assessing how well the top 20 recommended items matched user interests (Wang et al., 2019).

**Implementation Details.** We partitioned each dataset's global interaction graph into four subgraphs using spectral clustering (Damle et al., 2019) to align with the federated recommender systems setting (Table 1, 3, 9, 10, and Figure 3). Results were averaged over six runs across two different partitions. Random graphs were generated using the GNMK (Knuth, 1977) model, which effectively preserves degree distribution compared to the Erdos-Renyi (ER) model (Erdős et al., 1960). Further details on dataset statistics, baseline models, clustering methods, random graph generation, and all hyperparameter settings are provided in Appendix D, respectively.

## 4.2 EXPERIMENTAL RESULTS AND ANALYSIS

**(RQ1) Overall Performance Comparison.** Table 1 summarizes our comparative analysis against key graph-based FL baselines. We evaluate our framework in two settings: LPSFed(BPR), which applies a standard BPR loss (Rendle et al., 2009), and our full model LPSFed, which applies a BC loss (Eq. 9). As shown, LPSFed consistently outperforms all nine competing baselines, achieving state-of-the-art performance. LPSFed(BPR), leveraging our spectral personalization, already demonstrates strong performance by yielding robust personalized similarity measurements. LPSFed further incorporates our proposed bias-aware margin to mitigate local popularity bias. This synergy robustly handles both *structural imbalance* and *localized popularity bias*, delivering the significant NDCG gains (e.g., +24.2% in *Amazon-Book*) through more precise top-ranked item recommendations.

**(RQ2) Robustness to Subgraph Structural Imbalance.** Table 2 quantifies how well each method copes with *subgraph structural imbalance* on the *Amazon-Book*. The 15 clients are partitioned into three groups: **L**arge-**D**ense (**LD**, # Nodes > 40K, Avg. Degree 42.3), **M**edium-**B**alanced (**MB**, 10K < # Nodes < 15K, Avg. Degree 20.4), and **S**mall-**S**parse (**SS**, # Nodes < 8K, Avg. Degree 12.8). For each group, we report mean Recall@20 and NDCG@20; the last column averages over all clients.

**Table 2:** Performance for measuring the impact of subgraph structural imbalance on the *Amazon-Book* dataset, evaluated using Recall@20 and NDCG@20 across a 15-client partition. Clients are grouped into three categories: **L**arge-**D**ense, **M**edium-**B**alanced, **S**mall-**S**parse, and Overall (averaged across all clients). The best scores are highlighted in **bold** and the second-best in underlined. The *improvement* row shows the gains achieved by our model (**LPSFed**) compared to the second-best-performing model.

| Group | Large-Dense | | Medium-Balanced | | Small-Sparse | | Overall | |
|---|---|---|---|---|---|---|---|---|
| Model | Recall | NDCG | Recall | NDCG | Recall | NDCG | Recall | NDCG |
| FedAvg | 0.0620 | 0.0280 | 0.0411 | 0.0208 | 0.0275 | 0.0117 | 0.0334 | 0.0152 |
| FedPUB | 0.0605 | 0.0355 | 0.0528 | 0.0267 | 0.0300 | 0.0123 | 0.0381 | 0.0177 |
| FedMF | 0.0176 | 0.0094 | 0.0127 | 0.0056 | 0.0058 | 0.0027 | 0.0084 | 0.0039 |
| F2MF | 0.0402 | 0.0231 | 0.0341 | 0.0174 | 0.0225 | 0.0085 | 0.0268 | 0.0118 |
| PFedRec | 0.0619 | 0.0250 | 0.0525 | 0.0185 | 0.0310 | 0.0131 | 0.0388 | 0.0153 |
| FedRAP | 0.0073 | 0.0082 | 0.0067 | 0.0072 | 0.0048 | 0.0065 | 0.0055 | 0.0068 |
| FedPerGNN | 0.0053 | 0.0037 | 0.0038 | 0.0018 | 0.0039 | 0.0012 | 0.0040 | 0.0015 |
| FedHGNN | 0.0630 | 0.0319 | 0.0455 | 0.0237 | 0.0283 | 0.0095 | 0.0352 | 0.0148 |
| FedSSP | 0.0669 | 0.0380 | 0.0524 | 0.0269 | 0.0317 | 0.0126 | 0.0396 | 0.0181 |
| LPSFed (BPR) | 0.0655 | 0.0375 | 0.0523 | 0.0269 | 0.0306 | 0.0131 | 0.0387 | 0.0184 |
| **LPSFed** | **0.0769** | **0.0448** | **0.0550** | **0.0295** | **0.0331** | **0.0146** | **0.0419** | **0.0206** |
| Improvement | ↑ 14.9% | ↑ 17.9% | ↑ 4.2% | ↑ 9.7% | ↑ 4.4% | ↑ 11.5% | ↑ 5.8% | ↑ 13.8% |

**Table 3:** Comparison of the localized popularity bias mitigation on the *Amazon-Book* using NDCG@20. Best scores are in **bold**. Balanced data includes all data, while the imbalanced dataset excludes less active participants. Arrows show (%) change relative to FedAvg: ↑ gain (%), ↓ loss (%).

| Data Setting | Balanced Dataset [NDCG@20] | | | | Imbalanced Dataset [NDCG@20] | | | |
|---|---|---|---|---|---|---|---|---|
| Model | Tail | Mid | Head | Overall | Tail | Mid | Head | Overall |
| FedAvg | 0.0017 | 0.0077 | 0.0570 | 0.0264 | 0.0040 | 0.0108 | 0.0924 | 0.0312 |
| FedPUB | 0.0024 ↑41 | 0.008 ↑16 | 0.0620 ↑9 | 0.0279 ↑6 | 0.0064 ↑60 | 0.0130 ↑20 | 0.0998 ↑8 | 0.0322 ↑3 |
| FedMF | 0.0002 ↓88 | 0.0009 ↓88 | 0.0130 ↓77 | 0.0074 ↓72 | 0.0003 ↓92 | 0.0013 ↓88 | 0.0140 ↓85 | 0.0072 ↓77 |
| F2MF | 0.0037 ↑118 | 0.0112 ↑45 | 0.0438 ↓23 | 0.0225 ↓15 | 0.0050 ↑25 | 0.0119 ↑10 | 0.0436 ↓53 | 0.0225 ↓28 |
| PFedRec | 0.0055 ↑224 | 0.0125 ↑62 | 0.0277 ↓51 | 0.0244 ↓8 | 0.0068 ↑70 | 0.0145 ↑34 | 0.0262 ↓72 | 0.0242 ↓22 |
| FedRAP | 0.0031 ↑82 | 0.0097 ↑26 | 0.0098 ↓83 | 0.0110 ↓58 | 0.0014 ↓65 | 0.0055 ↓49 | 0.0107 ↓88 | 0.0090 ↓71 |
| FedPerGNN | 0.0002 ↓88 | 0.0007 ↓91 | 0.0042 ↓93 | 0.0015 ↓94 | 0.0003 ↓92 | 0.0008 ↓93 | 0.0046 ↓95 | 0.0026 ↓92 |
| FedHGNN | 0.0019 ↑12 | 0.0088 ↑14 | 0.0555 ↓3 | 0.0262 ↓1 | 0.0058 ↑45 | 0.0138 ↑28 | 0.0770 ↓17 | 0.0298 ↓4 |
| FedSSP | 0.0038 ↑124 | 0.0117 ↑52 | 0.0640 ↑12 | 0.0317 ↑20 | 0.0067 ↑68 | 0.0167 ↑55 | 0.1019 ↑10 | 0.0356 ↑14 |
| LPSFed (BPR) | 0.0031 ↑82 | 0.0113 ↑47 | 0.0610 ↑7 | 0.0301 ↑14 | 0.0068 ↑70 | 0.0149 ↑38 | 0.0974 ↑5 | 0.0322 ↑3 |
| **LPSFed** | **0.0063 ↑271** | **0.0143 ↑86** | **0.0752 ↑32** | **0.0390 ↑48** | **0.0078 ↑95** | **0.0193 ↑79** | **0.1052 ↑14** | **0.0442 ↑42** |

LPSFed (BPR), which leverages only spectral personalization, shows competitive performance, although its scores are slightly lower than the FedSSP. However, adding the bias-aware margin (LPSFed) yields significant performance gains across all groups and demonstrates three distinct effects. In group **LD**, it breaks the feedback loop, improving performance significantly over all baselines (+14.9% Recall, +17.9% NDCG over the best baseline). In group **MB**, the spectral-signal similarity effectively moderates update weights, allowing clients to benefit from the *global bias context* without overfitting to popular items. In group **SS**, the margin reduces over-dependence on a few popular items, enhancing long-tail recommendation quality.

**(RQ3) Localized Popularity Bias Mitigation.** Table 3 reports NDCG@20 on *Amazon-Book* under two settings: **Balanced** uses the full dataset, while **Imbalanced** excludes users and items with fewer than eight interactions. Users are grouped into **Tail**, **Mid**, and **Head** based on interaction proportions (3:2:1 ratio). We compare against FedAvg, and arrows indicate each model's relative change in NDCG@20. Traditional FL models like FedMF and FedPerGNN perform poorly in the Tail group, indicating that they are significantly affected by popularity bias. While PFedRec and FedSSP show notable gains in the Tail, they underperform our model (LPSFed) in the head. Similarly, F2MF, which uses auxiliary features for group-based fairness, performs better than LPSFed (BPR) in the Tail and mid categories on the balanced data, but underperforms in the overall categories on the Imbalanced data. However, LPSFed consistently achieves the best overall performance on both Balanced and Imbalanced datasets, indicating that our model's bias-aware margin removes popularity noise and preserves accuracy across the entire item spectrum.

**(RQ4) Ablation Study of Model Components.** Table 4 evaluates the contribution of each key component. The **w/o LGCN** variant, where we replace our core low-pass architecture with a standard

**Table 4:** Ablation study of LPSFed on five datasets, reporting Recall@20 and NDCG@20 for different model variants. **Bold** indicates best performance per column, and the last row shows relative *improvements* over the second-best.

| Dataset | Amazon-Book | | Gowalla | | Movielens-1M | | Yelp2018 | | Tmall-Buy | |
|---|---|---|---|---|---|---|---|---|---|---|
| Model | Recall | NDCG | Recall | NDCG | Recall | NDCG | Recall | NDCG | Recall | NDCG |
| w/o LGCN | 0.0388 | 0.0216 | 0.0965 | 0.0427 | 0.1622 | 0.0893 | 0.0471 | 0.0207 | 0.0256 | 0.0121 |
| w/o bias-aware | 0.0643 | 0.0322 | 0.1529 | 0.0711 | 0.2604 | 0.1281 | 0.0769 | 0.0301 | 0.0362 | 0.0186 |
| w/o per | 0.0652 | 0.0368 | 0.1576 | 0.0788 | 0.2625 | 0.1299 | 0.0775 | 0.0315 | 0.0370 | 0.0190 |
| w/o per & bias-aware | 0.0642 | 0.0312 | 0.1425 | 0.0660 | 0.2454 | 0.1240 | 0.0712 | 0.0292 | 0.0317 | 0.0164 |
| w/o statistics for $G_R$ | 0.0714 | 0.0403 | 0.1581 | 0.0856 | 0.2625 | 0.1330 | 0.0764 | 0.0372 | 0.0393 | 0.0199 |
| **LPSFed** | **0.0738** | **0.0442** | **0.1621** | **0.0909** | **0.2646** | **0.1342** | **0.0783** | **0.0379** | **0.0419** | **0.0206** |
| Improvement | ↑3.4% | ↑9.7% | ↑2.5% | ↑6.2% | ↑0.8% | ↑0.9% | ↑1.0% | ↑1.9% | ↑6.6% | ↑3.5% |

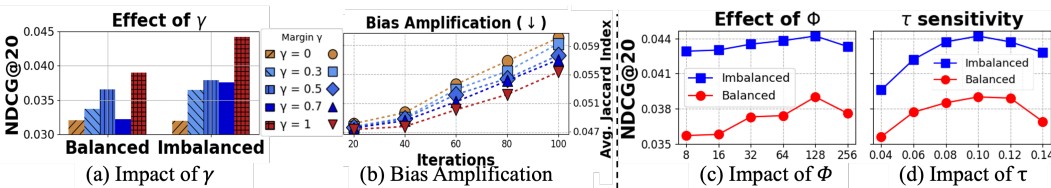

(a) Impact of γ    (b) Bias Amplification    (c) Impact of Φ    (d) Impact of τ

**Figure 3:** Hyperparameter sensitivity on the *Amazon-Book* dataset: (a) Bias-aware margin strength $\gamma$; (b) Impact of $\gamma$ on Bias Amplification in Imbalanced set; (c) Low-pass cut-off frequency $\Phi$; (d) Loss Temperature $\tau$.

spatial GNN (NGCF (Wang et al., 2019)), results in a catastrophic performance drop, which highlights the inherent limitations of spatial-based methods in handling structural divergence. Removing the personalization component, denoted **w/o per**, or the bias-aware margin, denoted **w/o bias-aware**, significantly degrades performance, confirming both modules are essential. Notably, **w/o statistics for** $G_R$ variant enforces stricter privacy by not aggregating any client statistics. Despite this, it performs robustly, achieving the second-best results across most metrics. Only the full LPSFed achieves consistently high Recall and precise rankings, confirming that these components enhance both broad coverage and accurate prediction.

**(RQ5) Hyperparameter Analysis.** Figure 3 shows how key hyperparameters affect NDCG@20 on *Amazon-Book* under **Balanced** and **Imbalanced** settings. **(a)** Increasing the bias-aware margin strength $\gamma$ consistently raises NDCG, as a stronger margin curbs the localized popularity bias and supports long-tail items. This indicates that maximizing the margin strength is the most effective strategy for achieving the intended bias mitigation objective. **(b)** Higher $\gamma$ suppresses the feedback loop by reducing recommendation overlap, thereby limiting popularity amplification. Specifically, $\gamma = 1.0$ yields the slowest growth in the Jaccard index, confirming that a stronger margin most effectively disrupts the popularity-driven feedback loop. **(c)** For the low-pass cut-off frequency $\Phi$, performance improves as more informative structural signals are captured, peaking at $\Phi = 128$, where the cut-off preserves the most spectrally stable frequencies while excluding high-frequency noise that would otherwise reduce robustness. **(d)** The BC-Loss temperature $\tau$ shows a similar pattern: appropriate settings strike a balance between stable optimization and sufficient exploration, while values that are too low hinder learning and those that are too high result in overfitting, both leading to lower NDCG.

## 5 CONCLUSION

In this paper, we introduce Low-pass Personalized Subgraph Federated Recommendation (LPSFed), a robust personalized FRS that simultaneously addresses *subgraph structural imbalance* and *localized popularity bias*. Our approach leverages low-pass spectral filtering for stable personalization, while a bias-aware margin mitigates feedback loops and improves long-tail recommendations. We provide theoretical justification for this framework, demonstrating that our similarity metric and our spectral filtering method are both governed by the same underlying eigengap, which validates our personalization strategy. Empirical evaluation on five real-world datasets confirms that this synergistic framework achieves state-of-the-art performance and robustness, outperforming all existing baselines. Further discussions on the limitations and LLM usage are provided in Appendix H and I.

## ACKNOWLEDGEMENTS

This work was supported by the Institute of Information and Communications Technology Planning and Evaluation (IITP), the Korea Creative Content Agency (KOCCA), and the National Research Foundation of Korea (NRF) through grants funded by the Korean government (RS-2024-00438686, RS-2024-00436936, IITP-2026-RS-2020-II201821, RS-2019-II190421, IITP-2025-RS-2024-00360227, RS-2025-02218768, RS-2025-25442569, RS-2025-24803185).

## ETHICS STATEMENT

This research was conducted in accordance with the ICLR Code of Ethics. The focus of our work is the development of robust, privacy-preserving federated recommender systems.

**Privacy and Security.** A primary ethical consideration in recommender systems is the handling of sensitive user data. Our federated learning framework directly addresses this challenge. By training models on decentralized, client-level subgraphs without sharing raw user-item interaction data, our method significantly enhances user privacy and reduces the risk of data exposure inherent in traditional centralized systems. The information shared with the server is limited to model parameters and minimal, non-invertible scalar values (e.g., structural similarity scores), which are designed to prevent the reconstruction of individual user data or local graph structures.

**Fairness, Bias, and Discrimination.** Recommender Systems can perpetuate and amplify existing biases, leading to unfair outcomes. Our research directly confronts this issue by proposing a method to mitigate localized popularity bias. By improving the quality of recommendations for diverse and structurally varied client groups, our work aims to promote fairness and provide more equitable exposure for long-tail items. This helps to counteract the feedback loops that often lead to a narrow, popularity-driven recommendation landscape, thereby fostering a more diverse and inclusive ecosystem.

**Data Usage and Research Integrity.** All experiments in this paper were conducted using publicly available and widely adopted benchmark datasets. These datasets consist of anonymized user interactions, and our work did not involve the collection of new data from human subjects. We are committed to research integrity and have provided detailed descriptions of our methodology and experimental setup to ensure reproducibility.

## REPRODUCIBILITY STATEMENT

To ensure the reproducibility of our work, we provide our source code as supplementary material. Our theoretical analysis is presented in Section 3.4, with detailed proofs for all theorems and supporting lemmas provided in Appendix B. Our code is available at: **`https://github.com/dntjr41/LPSFed`**. All experimental settings are described in Section 4.1 and Appendix D, which includes dataset statistics, the client subgraph clustering methodology, baseline model details, and a complete list of hyperparameters.

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

APPENDIX GUIDE

The appendix includes the theoretical analysis, related work, experimental setup, training efficiency, complexity analysis, additional experimental results, broader impacts, limitations, and LLM Usage. This guide provides a concise overview of its sections for easy navigation.

## A NOTATIONS

- **Table of Notations A**: List all key notations and symbols used in the paper.

## B THEORETICAL ANALYSIS B

- **Lemma 1 (Low-pass Filter Preservation)**: Shows low-pass filters preserve graph community structures with an eigengap-based error bound.
- **Lemma 2 (Spectral Measure Convergence)**: Proves eigenvalue distribution convergence in large graphs.
- **Lemma 3 (Filter Stability)**: Establishes low-pass filter stability under Laplacian perturbations.
- **Assumption 1 (Idealized Distribution)** Formalizes the idealized low-pass spectral distributions.
- **Theorem 3.1 (Structural Comparison)**: Compares graph structures via KL-divergence of eigenvalue distributions.
- **Theorem 3.2 (Spectral Regularization)**: Bounds variance of filtered node embeddings for community alignment.

## C RELATED WORK

- **Federated Recommender Systems C.1**: Examines collaborative filtering and personalized recommendation methods in federated settings.
- **Personalized Federated Learning C.2**: Covers approaches for handling data heterogeneity across clients, particularly in vision and graph domains.
- **Popularity Bias-aware Recommender Systems C.3**: Discusses methods for mitigating popularity bias in centralized recommendation settings.

## D DATASETS AND EXPERIMENTAL SETUP

- **Datasets D.1**: Details Movielens-1M, Gowalla, Yelp2018, Amazon-Book, Tmall-Buy datasets (Table 6).
- **Subgraph Clustering D.2**: Describes clustering methods for generating client subgraphs in the federated setting (Table 7).
- **Random Graph D.3**: Describes random graph generation in a federated setting.
- **Random Graph Sensitivity D.4**: Random Graph Sensitivity Analysis (Table 8).
- **Baselines D.5**: Lists baseline models for comparison.
- **Hyperparameter Settings D.6**: Summarizes configuration settings.

## E TRAINING EFFICIENCY & COMPLEXITY

- **Model Complexity E**: Demonstrates LPSFed achieves higher efficiency than state-of-the-art baselines.
- **Time Complexity**:
  - **Low-pass Convolution E.1.1**: Discusses efficient low-pass filtering and graph convolution operations.
  - **LPSFed Framework E.1.2**: Analyzes the overall complexity of the federated learning system.
- **Scalability E.1.2**: Validates the framework's support for large-scale graphs structures.
- **Costs E.1.2**: Assesses communication and computation overhead (Table 9).

## F  ADDITIONAL EXPERIMENTAL RESULTS

- **Bias Amplification Measurement and Effectiveness of the Bias-Aware Margin F.1**: Shows bias mitigation via Jaccard index, with higher margin strengths reducing convergence (Table 10).
- **Correlation Analysis Between Eigengap and Performance F.2**: Analyzes correlation, confirming spectral stability benefits within an appropriate filtering range (Table 11).
- **Effect of Client Variability on Performance F.3**: Evaluates performance across client counts (4, 10, 20), proving robustness to heterogeneity (Table 12).

## G  BROADER IMPACTS

- **Broader Impacts G**: Summarizes societal benefits such as privacy, fairness, and collaborative learning from our subgraph-level federated recommender systems approach.

## H  LIMITATIONS

- **Limitations H**: Discusses the limitations of this paper.

## I  LLM USAGE

- **LLM Usage I**: Details the use of an LLM for manusctipt preparation.

**Table 5:** Table of Notations.

| Symbol | Description |
|--------|-------------|
| *General Notations* | |
| $C$ | The total number of clients |
| $c$ | An index for a specific client, $c \in \{1, ..., C\}$ |
| $u, i$ | A user and an item, respectively |
| $U, I$ | The sets of all users and items in a given graph |
| $M, N$ | The number of users and items in a given graph |
| $D$ | The Dimensionality of embeddings |
| $\theta$ | General model parameters |
| *Graph Representation* | |
| $G, G_c$ | A global graph and a client's local subgraph |
| $\mathcal{V}, \mathcal{E}$ | The sets of nodes and edges in a graph |
| $\mathbf{A}, \mathbf{D}, \mathbf{L}$ | The adjacency, Degree, and Laplacian matrices of a graph |
| $\mathbf{R}$ | The user-item interaction matrix |
| $\lambda_i, \Lambda$ | The $i$-th eigenvalue and the diagonal matrix of eigenvalues |
| $\mathbf{P}$ | The matrix of eigenvectors of the Laplacian |
| *Low-pass Graph Convolutional Network* | |
| $\mathbf{Z}, \mathbf{U}, \mathbf{V}$ | The node, user, and item embedding matrices |
| $\mathbf{Z}^{(l)}$ | The node embedding matrix at the $l$-th layer |
| $\Phi$ | The cut-off frequency for the low-pass filter |
| $\bar{\mathbf{P}}$ | The truncated matrix of the first $\Phi$ eigenvectors |
| $\tilde{\mathbf{k}}, \bar{\mathbf{k}}$ | A convolution kernel in the frequency domain and its truncated version |
| $f_{\theta_{Pool}}, f_{\theta_{Pred}}$ | The pooling and Predictive MLPs |
| *Bias Mitigation* | |
| $p_u, p_i$ | Popularity scores for user $u$ and item $i$ |
| $f_{\psi_{bias}}, f_{\phi_{bias}}$ | Encoders for user and item popularity bias |
| $\hat{\xi}_{ui}$ | The angle between user and item popularity bias embeddings |
| $\mathcal{L}_{bias}, \mathcal{L}_{BC}$ | The auxiliary bias loss and the main Bias-aware Contrastive (BC) loss |
| $\mathcal{M}_{ui}^c$ | A locally computed adaptive margin for user $u$, item $i$ on client $c$ |
| $\widetilde{\mathcal{M}}_{ui}^c$ | The refined margin after interpolation with the global context |
| $\gamma, \tau, \omega$ | Hyperparameters: margin strength, softmax temperature, and interpolation weight |
| *Federated Learning (Low-pass Personalized Subgraph Federated Recommendation)* | |
| $\mathbf{K}^c, \mathbf{K}^R$ | The low-pass convolution kernel distributions for client $c$ and the reference graph $G_R$ |
| $\rho_c$ | The structural divergence of client $c$ (KL-divergence) |
| $\bar{\rho}_c$ | The normalized similarity score of client $c$ |
| $\mathcal{M}^c, \bar{\mathcal{M}}$ | A client $c$'s average margin and the aggregated global average margin |
| $\theta_{updated}^c$ | Updated (personalized) parameters for client $c$ |
| $\mathcal{M}_{updated}^c$ | The updated (personalized) global margin for client $c$ |
| *Theoretical Analysis* | |
| $\delta$ | The eigengap of the graph Laplacian ($\lambda_{k+1} - \lambda_k$) |
| $D_{struct}$ | The ideal structural divergence between theoretical $k$-block graphs |
| $\mathrm{Var}(\mathbf{z})$ | The graph smoothness (Dirichlet energy) of an embedding $\mathbf{z}$ |
| $\mathbf{z}^*$ | The theoretical $k$-community subspace embedding |

## A  NOTATIONS

Table 5 summarizes the key notations used throughout the paper.

## B  THEORETICAL ANALYSIS

**Lemma 1** (Low-pass Filter Preservation). *Let $G$ be an undirected graph with normalized Laplacian $\mathbf{L} = \mathbf{I} - \mathbf{D}^{-1/2}\mathbf{A}\mathbf{D}^{-1/2}$ and $k$ communities. For a low-pass filter $h_{\lambda_c}(\mathbf{L})$ with cut-off frequency $\lambda_c$ between $\lambda_k$ and $\lambda_{k+1}$:*

$$\|h_{\lambda_c}(\mathbf{L})\mathbf{x} - \Pi_{community}\mathbf{x}\|_2 \leq \frac{8\sqrt{k}}{\lambda_{k+1} - \lambda_k}\|\mathbf{x}\|_2, \tag{20}$$

*where $\Pi_{community}$ is the projection onto community indicator vectors.*

*Proof.* Let $\mathbf{L} = \mathbf{P}\mathbf{\Lambda}\mathbf{P}^{\mathsf{T}}$ be the eigendecomposition of $\mathbf{L}$, where $\mathbf{\Lambda} = \text{diag}(\lambda_1, \ldots, \lambda_n)$ with $0 = \lambda_1 \le \lambda_2 \le \cdots \le \lambda_n \le 2$. The proof proceeds in four steps:

First, let $\chi \in \mathbb{R}^{n \times k}$ be the matrix of true community indicators. By the variational characterization of eigenvalues, the first $k$ eigenvectors minimize:

$$\min_{\mathbf{X}^{\mathsf{T}}\mathbf{X}=\mathbf{I}_k} \text{tr}(\mathbf{X}^{\mathsf{T}}\mathbf{L}\mathbf{X}) = \sum_{i=1}^{k} \lambda_i. \tag{21}$$

Second, let $\mathbf{P}_k$ be the matrix of first $k$ eigenvectors. By the Davis-Kahan theorem (Davis & Kahan, 1970), when the eigengap $\lambda_{k+1} - \lambda_k > 0$:

$$\|\mathbf{P}_k - \chi\mathbf{R}\|_F \le \frac{8\sqrt{k}}{\lambda_{k+1} - \lambda_k}, \tag{22}$$

where $\mathbf{R}$ is an orthogonal matrix that best aligns $\mathbf{P}_k$ with $\chi$. Then, the low-pass filter $h_{\lambda_c}(\mathbf{L})$ with $\lambda_c \in (\lambda_k, \lambda_{k+1})$ acts as:

$$h_{\lambda_c}(\mathbf{L}) = \sum_{i=1}^{k} \mathbf{p}_i\mathbf{p}_i^{\mathsf{T}} = \mathbf{P}_k\mathbf{P}_k^{\mathsf{T}}. \tag{23}$$

Lastly, for any signal $\mathbf{x}$, using the projection $\Pi_{community} = \chi\chi^{\mathsf{T}}$:

$$\|h_{\lambda_c}(\mathbf{L})\mathbf{x} - \Pi_{community}\mathbf{x}\|_2 = \|\mathbf{P}_k\mathbf{P}_k^{\mathsf{T}}\mathbf{x} - \chi\chi^{\mathsf{T}}\mathbf{x}\|_2 \tag{24}$$

$$= \|(\mathbf{P}_k\mathbf{P}_k^{\mathsf{T}} - \chi\mathbf{R}\mathbf{R}^{\mathsf{T}}\chi^{\mathsf{T}})\mathbf{x}\|_2 \tag{25}$$

$$\le \|\mathbf{P}_k - \chi\mathbf{R}\|_F\|\mathbf{x}\|_2 \tag{26}$$

$$\le \frac{8\sqrt{k}}{\lambda_{k+1} - \lambda_k}\|\mathbf{x}\|_2. \tag{27}$$

$\square$

**Lemma 2** (Spectral Measure Convergence). *Let $G_n$ be a graph with $n$ vertices and a normalized low-pass filtered eigenvalue distribution with cut-off frequency $\lambda_c$. Let $\Phi$ $(< n)$ denote the number of eigenvalues below $\lambda_c$. For this distribution:*

$$\mathbf{K}^n(i) = \frac{\lambda_i^{(n)}}{\sum_{q=1}^{\Phi} \lambda_q^{(n)}}, \quad q \le \Phi. \tag{28}$$

*Then there exists a limiting distribution $\mathbf{K}^*$ such that:*

$$\mathbb{P}(\|\mathbf{K}^n - \mathbf{K}^*\|_{\infty} > \epsilon) \le 2\Phi \exp(-cn\epsilon^2), \tag{29}$$

*where $c, \epsilon > 0$ are constants.*

*Proof.* The proof proceeds in steps: By the Matrix Bernstein inequality (Tropp et al., 2015) for normalized Laplacians:

$$\mathbb{P}(\|\mathbf{L}^n - \mathbb{E}[\mathbf{L}^n]\| \ge t) \le 2n \exp(-\frac{nt^2}{4}). \tag{30}$$

Second, let $\mathbf{L}^* = \mathbb{E}[\mathbf{L}^n]$ be the limiting operator. For eigenvalues, Weyl's inequality gives:

$$|\lambda_i^{(n)} - \lambda_i^*| \le \|\mathbf{L}^n - \mathbf{L}^*\|_2. \tag{31}$$

For the normalized distribution $\mathbf{K}_n$:

$$|\mathbf{K}^n(i) - \mathbf{K}^*(i)| = \left|\frac{\lambda_i^{(n)}}{\sum_{q=1}^{\Phi} \lambda_q^{(n)}} - \frac{\lambda_i^*}{\sum_{q=1}^{\Phi} \lambda_q^*}\right| \tag{32}$$

$$\le \frac{|\lambda_i^{(n)} - \lambda_i^*|}{\sum_{q=1}^{\Phi} \lambda_q^{(n)}} + \frac{\lambda_i^*}{(\sum_{q=1}^{\Phi} \lambda_q^{(n)})^2}\left|\sum_{q=1}^{\Phi}(\lambda_q^{(n)} - \lambda_q^*)\right|. \tag{33}$$

Since eigenvalues of normalized Laplacians lie in $[0, 2]$ and $\Phi$ is fixed:

$$\sum_{q=1}^{\Phi} \lambda_q^{(n)} \geq c_1 > 0. \tag{34}$$

Therefore, for any $\epsilon > 0$:

$$|\mathbf{K}^n(i) - \mathbf{K}^*(i)| \leq C_1 \|\mathbf{L}^n - \mathbf{L}^*\|_2. \tag{35}$$

By setting $t = \epsilon/C_1$ in Eq. 30 and applying the union bound over $i \leq \Phi$:

$$\mathbb{P}(\|\mathbf{K}^n - \mathbf{K}^*\|_\infty > \epsilon) \leq 2\Phi \exp(-cn\epsilon^2), \tag{36}$$

where $c = 1/(4C_1^2)$. $\qquad\square$

**Lemma 3** (Filter Stability). *For a low-pass filter $h_{\lambda_c}(\mathbf{L})$ and perturbed Laplacian $\tilde{\mathbf{L}} = \mathbf{L} + \mathbf{E}$:*

$$\|h_{\lambda_c}(\mathbf{L}) - h_{\lambda_c}(\tilde{\mathbf{L}})\|_2 \leq \frac{\|\mathbf{E}\|_2}{\delta_{\lambda_c}}, \tag{37}$$

*where $\delta_{\lambda_c}$ is the minimum gap between eigenvalues separated by $\lambda_c$.*

*Proof.* We prove this in three steps using complex analysis: First, express the filter difference using the Cauchy integral formula:

$$h_{\lambda_c}(\mathbf{L}) - h_{\lambda_c}(\tilde{\mathbf{L}}) = \frac{1}{2\pi i} \oint_\Gamma (\mathbf{w}\mathbf{I} - \mathbf{L})^{-1} \mathbf{E} (\mathbf{w}\mathbf{I} - \tilde{\mathbf{L}})^{-1} d\mathbf{w}, \tag{38}$$

where $\Gamma$ is a contour enclosing eigenvalues below $\lambda_c$.

For the resolvent norm, when $\mathbf{w}$ is on $\Gamma$:

$$\|(\mathbf{w}\mathbf{I} - \mathbf{L})^{-1}\|_2 \leq \frac{1}{\text{dist}(\mathbf{w}, \sigma(\mathbf{L}))} \leq \frac{1}{\delta_{\lambda_c}}, \tag{39}$$

where $\sigma(\mathbf{L})$ is the spectrum of $\mathbf{L}$. By taking operator norms:

$$\|h_{\lambda_c}(\mathbf{L}) - h_{\lambda_c}(\tilde{\mathbf{L}})\|_2 \tag{40}$$

$$\leq \frac{1}{2\pi} \oint_\Gamma \|(\mathbf{w}\mathbf{I} - \mathbf{L})^{-1}\|_2 \|\mathbf{E}\|_2 \|(\mathbf{w}\mathbf{I} - \tilde{\mathbf{L}})^{-1}\|_2 |d\mathbf{w}| \tag{41}$$

$$\leq \frac{\|\mathbf{E}\|_2}{\delta_{\lambda_c}}. \tag{42}$$

The final inequality uses the fact that the contour integral equals $2\pi i$ for the characteristic function of $(-\infty, \lambda_c]$. $\qquad\square$

**Assumption 1** (Idealized Low-Pass Spectral Distributions). *For each graph $j \in \{1, 2\}$, let $G_j = (\mathcal{V}_j, \mathcal{E}_j)$ be an undirected (possibly bipartite) graph with normalized Laplacian $\mathbf{L}^{(j)} = \mathbf{I} - \mathbf{D}^{(j)-\frac{1}{2}} \mathbf{A}^{(j)} \mathbf{D}^{(j)-\frac{1}{2}}$ and eigenvalues $0 = \lambda_1^{(j)} \leq \lambda_2^{(j)} \leq \cdots \leq \lambda_{n_j}^{(j)} \leq 2$. Fix a common low-pass cut-off $\lambda_c \in (0, 2)$ (for bipartite graphs choose $\lambda_c < 2$), and let $\Phi^{(j)} := \max\{m : \lambda_m^{(j)} \leq \lambda_c\}$. Define the empirical low-pass eigenvalue distribution:*

$$\mathbf{K}^{(j)} \in \Delta^{\Phi^{(j)}-1}, \qquad \mathbf{K}^{(j)}(i) = \frac{\lambda_i^{(j)}}{\sum_{q=1}^{\Phi^{(j)}} \lambda_q^{(j)}}, \quad i \leq \Phi^{(j)}. \tag{43}$$

*Idealized model.* We assume $G_j$ is generated by a $k$-community structural model that induces an idealized *(expected) graph:* either

(SBM) *a $k$-block (bi)SBM with community proportions $\alpha^{(j)}$ and block matrix $B^{(j)}$, with expected adjacency $A^{(j),\star} = \mathbb{E}[A^{(j)}]$; or*

(Graphon) *a piecewise-constant $k$-block graphon $W_j$, with associated integral operator's discretization $\boldsymbol{A}^{(j),\star}$.*

*Let $\mathbf{L}_{struct}^{(j)}$ be the normalized Laplacian of $\boldsymbol{A}^{(j),\star}$, with eigenvalues $0 = \lambda_{struct,1}^{(j)} \leq \cdots \leq \lambda_{struct,n_j}^{(j)}$ and $\Phi_{struct}^{(j)} := \max\{m : \lambda_{struct,m}^{(j)} \leq \lambda_c\}$. The* idealized *low-pass distribution is:*

$$\mathbf{K}_{struct}^{(j)}(i) = \frac{\lambda_{struct,i}^{(j)}}{\sum_{q=1}^{\Phi_{struct}^{(j)}} \lambda_{struct,q}^{(j)}}, \quad i \leq \Phi_{struct}^{(j)}. \tag{44}$$

***Structural divergence.*** *The population (idealized) structural divergence between two graphs $a, b$ is:*

$$D_{struct} := D_{\mathrm{KL}}\big(\mathbf{K}_{struct}^{(a)} \| \mathbf{K}_{struct}^{(b)}\big), \tag{45}$$

*where $D_{\mathrm{KL}}$ is computed after $\varepsilon$-smoothing of zero coordinates in the denominator (i.e., replace any zero $\mathbf{q}_i$ by $\max\{\mathbf{q}_i, \varepsilon\}$ for a fixed $\varepsilon > 0$).*

**Theorem 3.1** (Structural Comparison via Spectral Distributions). Let $G_1 = (\mathcal{V}_1, \mathcal{E}_1)$ and $G_2 = (\mathcal{V}_2, \mathcal{E}_2)$ be graphs with $n_1 = |\mathcal{V}_1|$ and $n_2 = |\mathcal{V}_2|$ nodes and $k$ communities each. $\mathcal{E}_1$ and $\mathcal{E}_2$ are sets of edges of $G_1$ and $G_2$, respectively. Moreover, $\Phi$ $(< n)$ denotes the number of eigenvalues below the cut-off frequency $\lambda$. Let $\mathbf{K}^1$ and $\mathbf{K}^2$ be their respective low-pass filtered eigenvalue distributions:

$$\mathbf{K}^j(i) = \frac{\lambda_i^{(j)}}{\sum_{q=1}^{\Phi} \lambda_q^{(j)}}, \quad q \leq \Phi, j \in \{1, 2\}. \tag{46}$$

Under Assumption 1, let $D_{\mathrm{struct}} = D_{KL}(\mathbf{K}_{\mathrm{struct}}^1 \| \mathbf{K}_{\mathrm{struct}}^2)$ denote the KL-divergence between the idealized distributions $\mathbf{K}_{\mathrm{struct}}^1$ and $\mathbf{K}_{\mathrm{struct}}^2$ corresponding to the $k$-community structures of $G_1$ and $G_2$, respectively. $c, \epsilon > 0$ are constants. Then, (1) The KL-divergence converges to a limiting value $D^*$ with probability:

$$\mathbb{P}(|D_{KL}(\mathbf{K}^1 \| \mathbf{K}^2) - D^*| > \epsilon) \leq 4\Phi \exp(-c \min(n_1, n_2)\epsilon^2/8). \tag{47}$$

(2) The structural similarity is preserved with error:

$$|D_{KL}(\mathbf{K}^1 \| \mathbf{K}^2) - D_{\mathrm{struct}}| \leq \frac{C}{\min(\delta_1, \delta_2)}, \tag{48}$$

where $C$ is a constant and $\delta_j$ is the eigengap $\lambda_{k+1}^{(j)} - \lambda_k^{(j)}$ for graph $G_j$.

*Proof.* For each graph $G_j$, by Lemma 2:

$$\mathbb{P}(\|\mathbf{K}^j - \mathbf{K}_*^j\|_\infty > \epsilon) \leq 2\Phi \exp(-cn_j\epsilon^2). \tag{49}$$

Meanwhile, for the KL-divergence, we decompose:

$$|D_{KL}(\mathbf{K}^1 \| \mathbf{K}^2) - D_{KL}(\mathbf{K}_*^1 \| \mathbf{K}_*^2)| \tag{50}$$

$$\leq |D_{KL}(\mathbf{K}^1 \| \mathbf{K}^2) - D_{KL}(\mathbf{K}_*^1 \| \mathbf{K}^2)| \tag{51}$$

$$+ |D_{KL}(\mathbf{K}_*^1 \| \mathbf{K}^2) - D_{KL}(\mathbf{K}_*^1 \| \mathbf{K}_*^2)|. \tag{52}$$

Then, since $\mathbf{K}^j$ are probability distributions bounded away from 0 (due to low-pass filtering), we can apply the Lipschitz property of KL-divergence:

$$|D_{KL}(\mathbf{K}^1 \| \mathbf{K}^2) - D_{KL}(\mathbf{K}_*^1 \| \mathbf{K}^2)| \tag{53}$$

$$\leq C(\|\mathbf{K}^1 - \mathbf{K}_*^1\|_\infty + \|\mathbf{K}^2 - \mathbf{K}_*^2\|_\infty). \tag{54}$$

For each graph, Lemma 2 gives the probability bounds for deviations:

$$\mathbb{P}(\|\mathbf{K}^1 - \mathbf{K}_*^1\|_\infty > \epsilon) \leq 2\Phi \exp(-cn_1\epsilon^2) \tag{55}$$

$$\mathbb{P}(\|\mathbf{K}^2 - \mathbf{K}_*^2\|_\infty > \epsilon) \leq 2\Phi \exp(-cn_2\epsilon^2). \tag{56}$$

By the union bound, the probability of either distribution deviating is bounded by:

$$\mathbb{P}(\|\mathbf{K}^1 - \mathbf{K}^1_*\|_\infty > \epsilon \text{ or } \|\mathbf{K}^2 - \mathbf{K}^2_*\|_\infty > \epsilon) \tag{57}$$

$$\leq 2\Phi\exp(-cn_1\epsilon^2) + 2\Phi\exp(-cn_2\epsilon^2) \tag{58}$$

$$\leq 2\Phi(\exp(-cn_1\epsilon^2) + \exp(-cn_2\epsilon^2)) \tag{59}$$

$$\leq 4\Phi\exp(-c\min(n_1, n_2)\epsilon^2). \tag{60}$$

The final bound uses the fact that:

$$\exp(-cn_1\epsilon^2) + \exp(-cn_2\epsilon^2) \leq 2\exp(-c\min(n_1, n_2)\epsilon^2). \tag{61}$$

For the structural preservation bound, we proceed in several steps: First, from Lemma 1, for each graph $G_j$, the low-pass filter preserves community structure with error:

$$\|h_{\lambda_c}(\mathbf{L}_j)\mathbf{x} - \Pi^{(j)}_{\text{community}}\mathbf{x}\|_2 \leq \frac{8\sqrt{k}}{\lambda^{(j)}_{k+1} - \lambda^{(j)}_k}\|\mathbf{x}\|_2 = \frac{8\sqrt{k}}{\delta_j}\|\mathbf{x}\|_2. \tag{62}$$

By Lemma 3, when the Laplacian is perturbed by $\mathbf{E}$, the filter output changes by at most:

$$\|h_{\lambda_c}(\mathbf{L}^j) - h_{\lambda_c}(\tilde{\mathbf{L}}^j)\|_2 \leq \frac{\|\mathbf{E}\|_2}{\delta_j}. \tag{63}$$

Similarly, this perturbation affects the filter output by:

$$\|h_{\lambda_c}(\mathbf{L}^j) - h_{\lambda_c}(\mathbf{L}^j_{\text{struct}})\|_2 \leq \frac{C_0}{\delta_j}. \tag{64}$$

For the eigenvalues, this implies:

$$|\lambda^{(j)}_i - \lambda^{\text{struct}}_i| \leq \frac{C_0}{\delta_j}. \tag{65}$$

Let $\mathbf{K}^j_{\text{struct}}$ be the distribution that perfectly captures the community structure. Then:

$$\|\mathbf{K}^j - \mathbf{K}^j_{\text{struct}}\|_\infty \leq \frac{C_1}{\delta_j}, \tag{66}$$

where $C_1$ depends on the number of communities $k$. For the KL-divergence between distributions:

$$|D_{KL}(\mathbf{K}^1\|\mathbf{K}^2) - D_{KL}(\mathbf{K}^1_{\text{struct}}\|\mathbf{K}^2_{\text{struct}})| \tag{67}$$

$$\leq C_2(\|\mathbf{K}^1 - \mathbf{K}^1_{\text{struct}}\|_\infty + \|\mathbf{K}^2 - \mathbf{K}^2_{\text{struct}}\|_\infty) \tag{68}$$

$$\leq C_2(\frac{C_1}{\delta_1} + \frac{C_1}{\delta_2}) \tag{69}$$

$$\leq \frac{2C_1C_2}{\min(\delta_1, \delta_2)}. \tag{70}$$

By defining $D_{\text{struct}} = D_{KL}(\mathbf{K}^1_{\text{struct}}\|\mathbf{K}^2_{\text{struct}})$ and letting $C = 2C_1C_2$:

$$|D_{KL}(\mathbf{K}^1\|\mathbf{K}^2) - D_{\text{struct}}| \leq \frac{C}{\min(\delta_1, \delta_2)}. \tag{71}$$

This bound shows that the KL-divergence between the filtered distributions approximates the true structural similarity up to an error controlled by the minimum eigengap of the two graphs. $\qquad\square$

**Theorem 3.2** (Spectral Regularization). Let $G = (\mathcal{V}, \mathcal{E})$ be a client graph with $n = |\mathcal{V}|$ and the eigengap $\delta = \lambda_{k+1} - \lambda_k > 0$, where $k$ is the number of communities in the graph $G$. Denote the low-pass filtered node embedding and the $k$-community subspace embeddings by $\mathbf{z}, \mathbf{z}^* \in \mathbb{R}^n$, respectively. Assume the raw user/item embeddings are bounded by $\|\mathbf{U}_u\|_2, \|\mathbf{V}_i\|_2 \leq r$, where $r > 0$ is the embedding $\ell_2$ norm bound.

$$| \operatorname{Var}(\mathbf{z}) - \operatorname{Var}(\mathbf{z}^*)| \leq \frac{C'}{\delta}, \tag{72}$$

where $\operatorname{Var}(\cdot)$ denotes graph smoothness and $C' = 32\sqrt{k}r^2$.

*Proof.* By Lemma 1 (Low-pass Filter Preservation), the raw embedding $\mathbf{x} \in \mathbb{R}^n$ represents the initial user/item embeddings with $\|\mathbf{x}\|_2 \leq r$. The $k$-community subspace embedding $\mathbf{z}^*$ reflects the community structure of the graph and is defined as:

$$\mathbf{z}^* = \Pi_{\text{community}}\mathbf{x}, \tag{73}$$

where $\Pi_{\text{community}}$ is the projection onto the community subspace, spanned by the smallest $k$ eigenvectors of the graph Laplacian $\mathbf{L}$. Since $\mathbf{z}^*$ is a projection of $\mathbf{x}$, we have $\|\mathbf{z}^*\|_2 \leq \|\mathbf{x}\|_2 \leq r$. This clarifies that $\mathbf{z}^*$ represents $\mathbf{x}$ projected into the low-frequency $k$-community subspace.

The Low-pass filtered embedding is given by $\mathbf{z} = h_{\lambda_c}(\mathbf{L})\mathbf{x}$, where $h_{\lambda_c}(\mathbf{L})$ is a low-pass filter that preserves components associated with small eigenvalues of $\mathbf{L}$ (low frequencies) and attenuates those with large eigenvalues (high frequencies). By Lemma 1, the approximation error is bounded as:

$$\|\mathbf{z} - \mathbf{z}^*\|_2 \leq \frac{8\sqrt{k}}{\delta}\|\mathbf{x}\|_2 \leq \frac{8\sqrt{k}}{\delta}r, \tag{74}$$

where $\delta = \lambda_{k+1} - \lambda_k$ is the eigengap. This shows that $\mathbf{z}$ approximates the embedding $\mathbf{z}^*$.

The Dirichlet energy difference is:

$$| \operatorname{Var}(\mathbf{z}) - \operatorname{Var}(\mathbf{z}^*)| = |\mathbf{z}^\top \mathbf{L}\mathbf{z} - (\mathbf{z}^*)^\top \mathbf{L}\mathbf{z}^*|. \tag{75}$$

Let $\Delta = \mathbf{z} - \mathbf{z}^*$, with $\|\Delta\|_2 \leq \frac{8\sqrt{k}}{\delta}r$.

Expanding the difference:

$$\mathbf{z}^\top \mathbf{L}\mathbf{z} = (\mathbf{z}^* + \Delta)^\top \mathbf{L}(\mathbf{z}^* + \Delta) = (\mathbf{z}^*)^\top \mathbf{L}\mathbf{z}^* + 2(\mathbf{z}^*)^\top \mathbf{L}\Delta + \Delta^\top \mathbf{L}\Delta, \tag{76}$$

so:

$$| \operatorname{Var}(\mathbf{z}) - \operatorname{Var}(\mathbf{z}^*)| = |2(\mathbf{z}^*)^\top \mathbf{L}\Delta + \Delta^\top \mathbf{L}\Delta|. \tag{77}$$

We bound each term:

- **Cross term**:

$$|(\mathbf{z}^*)^\top \mathbf{L}\Delta| \leq \|\mathbf{z}^*\|_2 \|\mathbf{L}\Delta\|_2 \leq \|\mathbf{z}^*\|_2 \cdot \|\mathbf{L}\|_2 \cdot \|\Delta\|_2. \tag{78}$$

Since $\|\mathbf{L}\|_2 \leq 2$ (for a normalized Laplacian), $\|\mathbf{z}^*\|_2 \leq r$, and $\|\Delta\|_2 \leq \frac{8\sqrt{k}}{\delta}r$,

$$|(\mathbf{z}^*)^\top \mathbf{L}\Delta| \leq r \cdot 2 \cdot \frac{8\sqrt{k}}{\delta}r = \frac{16\sqrt{k}r^2}{\delta}, \tag{79}$$

$$2|(\mathbf{z}^*)^\top \mathbf{L}\Delta| \leq \frac{32\sqrt{k}r^2}{\delta}. \tag{80}$$

- **Quadratic term**:

$$|\Delta^\top \mathbf{L}\Delta| \leq \|\mathbf{L}\|_2 \|\Delta\|_2^2 \leq 2\left(\frac{8\sqrt{k}}{\delta}r\right)^2 = \frac{128kr^2}{\delta^2}. \tag{81}$$

Combining these:

$$|\operatorname{Var}(\mathbf{z}) - \operatorname{Var}(\mathbf{z}^*)| \leq \frac{32\sqrt{k}r^2}{\delta} + \frac{128kr^2}{\delta^2}. \tag{82}$$

For large $\delta$, the $\frac{1}{\delta}$ term dominates, yielding:

$$|\operatorname{Var}(\mathbf{z}) - \operatorname{Var}(\mathbf{z}^*)| \leq \frac{C'}{\delta}, \quad C' = 32\sqrt{k}r^2. \tag{83}$$

$\square$

## C  RELATED WORK

### C.1  FEDERATED RECOMMENDER SYSTEMS

Federated Recommender Systems (FRS) enhance privacy by processing data locally while sharing only model updates. Methods like FedMF (Chai et al., 2020) and LightFR (Zhang et al., 2023b) use federated matrix factorization for global collaborative filtering without data leakage, while F2MF Liu et al. (2022) integrates user/item features for fairness. Additionally, for e-commerce platforms, (Park et al., 2026) introduces a data valuation framework to quantify the utility of decentralized data during the federation process. To improve recommendation relevance, recent research has shifted toward Personalized Federated Recommendations (PFR), which tailor predictions to individual client preferences. For instance, PFedRec (Zhang et al., 2023a) integrates dual personalization strategies, and FedRAP (Li et al., 2024) introduces adjustable personalized layers. Although PFR frameworks address personalization, many treat clients (users) independently, limiting their ability to capture high-order interactions inherent to Graph Neural Networks (GNNs). To alleviate this, existing methods have explored ways to incorporate relational structure. Notable examples include FedPerGNN (Wu et al., 2022), which expands subgraphs via a third-party server, FedHGNN (Yan et al., 2024) utilizes heterogeneous GNN to handle diverse data relationships, and SemiDEFGL (Qu et al., 2023) augments ego-graphs with synthetic common items. However, these approaches mainly focus on augmenting the local neighborhood. Furthermore, F2PGNN (Agrawal et al., 2024) addresses the fairness problem; it relies on additional feature information beyond user-item interactions. Importantly, neither leverage *graph spectral signals*, which capture fundamental structural patterns, nor explicitly address the *subgraph structural imbalance* caused by variations in client subgraph scales and connectivities.

### C.2  PERSONALIZED FEDERATED LEARNING

Personalized Federated Learning (PFL) extends standard FL by incorporating client-specific adaptations to handle heterogeneous data distributions. In vision tasks, heterogeneity is driven largely by disparities in local data sizes. For example, FedVC (Hsu et al., 2020) reweights and resamples clients to address data-size gaps, Astraea (Duan et al., 2020) employs augmentation and rescheduling to self-balance imbalanced datasets, and q-FFL (Li et al., 2020) uses a fairness loss to reweight underrepresented clients. In graph-based node classification tasks, heterogeneity arises from class imbalance and varying class-driven graph topology. FedPer (Arivazhagan et al., 2019) and FedSim (Palihawadana et al., 2022) address this by using adaptive layers and similarity-guided aggregation, while G-FML (Yang et al., 2023), FedGSL (Zhao et al., 2022), and FedCog (Lei et al., 2023) leverage subgraph augmentation and meta-learning to align diverse local graphs. Similarly, CUFL (Kang & Park, 2025) introduces a curriculum-guided strategy to adaptively personalize subgraph FL under data heterogeneity. Additionally, FedPUB (Baek et al., 2023) clusters clients based on random-graph similarity, and FedSSP (Tan et al., 2024) and S2FGL (Tan et al., 2025a) apply low-pass filters for graph and node classification. However, existing PFL methods for graph data have largely focused on these types of heterogeneity, whereas the core challenge in recommender systems is the structural imbalance of interaction-only subgraphs that vary widely in size and connectivity. As a result, they fail to address the *subgraph structural imbalance* characteristics of FRS, and are ineffective at mitigating structural divergence across clients.

### C.3  POPULARITY BIAS-AWARE RECOMMENDER SYSTEMS

GNN-based Recommender Systems, such as NGCF (Wang et al., 2019), DGCF (Wang et al., 2020b), and DHCF (Ji et al., 2020), have significantly improved recommendation accuracy in centralized

**Table 6:** The statistics of datasets.

| Dataset | Movielens-1M | Gowalla | Yelp2018 | Amazon-Book | Tmall-Buy |
|---|---|---|---|---|---|
| Number of Users | 6,040 | 29,858 | 31,668 | 52,643 | 885,759 |
| Number of Items | 3,900 | 40,981 | 38,048 | 91,599 | 1,144,124 |
| Number of Interactions | 1,000,290 | 1,027,370 | 1,561,406 | 2,984,108 | 7,592,214 |
| Density | 5.431% | 0.084% | 0.130% | 0.062% | 0.010% |

**Table 7:** Comparison of different client construction strategies on *Movielens-1M*.

| Method ($Avg. \pm std$) | Ego-graph | Random-Const. | Interconnected-Const. | Spectral-Clust. |
|---|---|---|---|---|
| Avg. Subgraph Size | $141.1 \pm 16.8$ | $172.8 \pm 9.8$ | $2105.6 \pm 135.7$ | $196.2 \pm 36.2$ |
| Size variance | $22276.8 \pm 6905.9$ | $3198.5 \pm 1039.8$ | $85083.4 \pm 45597.3$ | $33370.3 \pm 17210.8$ |
| Avg. Degree | $1.97 \pm 0.0$ | $5.68 \pm 0.24$ | $7.29 \pm 0.47$ | $7.57 \pm 1.07$ |
| Degree Variance | $1.0 \pm 0.0$ | $6.34 \pm 0.08$ | $2.19 \pm 1.08$ | $30.22 \pm 11.45$ |
| Avg. Density | $0.032 \pm 0.004$ | $0.035 \pm 0.001$ | $0.003 \pm 0.000$ | $0.070 \pm 0.022$ |
| Spectral Entropy | $1.0 \pm 0.0$ | $6.34 \pm 0.08$ | $6.42 \pm 0.15$ | $5.69 \pm 0.68$ |
| Avg. Low-Freq. Energy | $0.996 \pm 0.0$ | $0.365 \pm 0.010$ | $1.0 \pm 0.0$ | $0.357 \pm 0.030$ |
| Avg. Subgraphs/Iter. | 100 | 10 | 8 | 4 |

systems but often struggle with popularity bias. To address this, recent methods adopt advanced loss functions that mitigate bias and enhance fairness, including DirectAU (Wang et al., 2022), CausE (Bonner & Vasile, 2018), IPS (Saito et al., 2020), and BC-Loss (Zhang et al., 2022), which rely on global item-popularity statistics or direct embedding adjustments. However, in FL, such information is inaccessible, and sharing it would break privacy guarantees, as no client has access to the full interaction graph. Our approach introduces a privacy-preserving bias-aware margin by having each client compress its local popularity distributions skew into a single value, which the server aggregates into a *global bias context* to regularize local model updates. This process disrupts the popularity-driven feedback loop without compromising client privacy.

# D  EXPERIMENTAL SETTINGS

## D.1  DATASETS

We use five real-world datasets to evaluate the recommendation performance of our proposed method.

- *Movielens-1M (mov)* is people's expressed preferences for movies. These preferences are in the form of tuples, each showing a person's rating (0-5 stars) for a specific movie at a particular time.
- *Gowalla* (He et al., 2020) is a location-based social networking website where users share their locations by checking in. The friendship network is undirected and was collected using their public API.
- *Yelp2018 (yel)* is derived from the 2018 edition of the Yelp challenge. In this challenge, local businesses such as restaurants and bars are treated as items. Yelp maintains the same 10-core setting to ensure data quality.
- *Amazon-Book* (He et al., 2020) is used in Amazon-Review for product recommendation purposes.
- *Tmall-Buy* (Tma) is a large-scale, real-world dataset from the Tmall e-commerce platform. It is widely used for benchmarking the performance of commercial-scale recommender systems under conditions of extreme data sparsity.

## D.2  SUBGRAPH CLUSTERING

A key challenge in FRS research is the lack of public, partitioned benchmarks with structural imbalance. To create a rigorous and reproducible benchmark to evaluate robustness, we must simulate client partitions. We empirically compared four common partitioning strategies on the *Movielens-1M* dataset (results in Table 7) to find the one that generates the most realistic and challenging structural heterogeneity.

**Table 8:** Sensitivity analysis of the anchor graph design in LPSFed. Performance is reported using Recall@20 across five datasets. "w/o statistics for $G_R$ (GNMK)" constructs the GNMK reference solely from global assumptions, without any client-derived statistics. **Bold** indicates best performance per column.

| Anchor Design | Amazon-Book | Gowalla | ML-1M | Yelp2018 | Tmall-Buy |
|---|---|---|---|---|---|
| ER | 0.0734 | 0.1599 | 0.2631 | 0.0767 | 0.0410 |
| GNMK | **0.0738** | **0.1621** | **0.2646** | **0.0783** | **0.0419** |
| w/o statistics for $G_R$ (GNMK) | 0.0714 | 0.1581 | 0.2625 | 0.0764 | 0.0393 |

The strategies are: (1) **Ego-graph**: Each client is a 1-hop ego network of a single user. (2) **Random-Const.**: Each client is formed from a random subset of users. (3) **Interconnected-Const.**: Each client is a full bipartite graph induced by a group of users. (4) **Spectral-Clust.**: Our chosen method. As Table 7 shows, other methods yield structurally weak or homogeneous subgraphs (e.g., low degree variance). In contrast, **Spectral-Clust.** produces subgraphs with high internal connectivity and, crucially, the **highest degree variance** and diverse structural statistics. Therefore, we adopted spectral clustering (Damle et al., 2019) for our experiments, as it provides the most effective protocol to simulate the *subgraph structural imbalance* this paper aims to solve. For the main performance evaluation (Table 1), we applied spectral clustering to partition the entire user-item interaction graph into four distinct subgraphs, ensuring manageable complexity and clear separations. We also introduced diversity in edge distributions across these subgraphs (approx. $\pm 20\%$ variation) to ensure each captures different degrees of user-item engagement. Furthermore, to mitigate any bias from a single partitioning outcome, we repeated this spectral clustering process twice, and all reported results are the average of three runs per split (a total of six runs).

### D.3 RANDOM GRAPH

To construct random reference graphs, we used the average number of user nodes, item nodes, interactions, and mean degree per subgraph to generate two types of bipartite graphs: the Erdos-Renyi (ER) model (Erdős et al., 1960) or the GNMK model (Knuth, 1977). The ER model creates edges between node pairs with a fixed probability, simulating purely random user-item interactions. In contrast, the GNMK model constructs a bipartite graph with a specified number of edges, resulting in more structured connectivity.

Both ER and GNMK models were evaluated in our experiments. The ER model, due to its high randomness, struggles to reflect the structural properties of real-world user-item graphs. Its inherent randomness fails to represent community-like structures or degree imbalance, leading to suboptimal personalization similarity measurements. Moreover, the eigenvalue spectrum of ER graphs asymptotically converges to the free convolution of the Gaussian and Wigner semicircular distributions (Ramakrishna et al., 2020), exhibiting unclear spectral gaps. This absence of spectral separation limits the design of effective low-pass filters, further reducing the model's utility. In contrast, the GNMK model, which generates graphs with a fixed number of edges, better preserves the structural patterns and degree distribution of clustered subgraphs. This closer alignment with real-world data characteristics resulted in the GNMK model consistently outperforming the ER model, delivering improved performance across all metrics.

### D.4 SENSITIVITY TO ANCHOR GRAPH DESIGN

To evaluate the impact of the reference graph used for structural comparison, we conduct an ablation study using two widely adopted random graph models: Erdos-Renyi (ER) and the more structure-aware GNMK model. As shown in Table 8, the GNMK-based anchor consistently achieves the best performance across all datasets, confirming our motivation that GNMK better captures the structural properties of real-world recommendation graphs. Importantly, we also evaluate a stricter privacy setting "w/o statistics for $G_R$", where the GNMK reference graph is constructed without using any client-derived statistics. This variant yields performance close to the full model, demonstrating that LPSFed remains robust even without access to client-specific structural information. These findings validate the effectiveness of the GNMK anchor while demonstrating that our framework maintains stability under stronger privacy constraints.

## D.5 BASELINES

We evaluate our proposed method in comparison to nine baselines:

- **FedAvg** (McMahan et al., 2017): is a foundational federated learning approach that transmits parameter gradients instead of data, enhancing privacy.
- **FedPUB** (Baek et al., 2023): adjusts weights within community structures using random graphs and parameter masking to protect data privacy.
- **FedMF** (Chai et al., 2020): applies matrix factorization within federated settings to ensure secure and private collaborative filtering.
- **F2MF** (Liu et al., 2022): aims to address fairness issues in federated recommender systems by incorporating fairness constraints into the matrix factorization process, ensuring equitable recommendations across diverse user groups.
- **PFedRec** (Zhang et al., 2023a): offers a lightweight, user-specific model for on-device personalization without user embeddings.
- **FedRAP** (Li et al., 2024): utilizes a dual personalized approach to manage user and item embeddings separately, optimizing communication efficiency.
- **FedPerGNN** (Wu et al., 2022): creates subgraphs for users with shared interactions, utilizing Local Differential Privacy (LDP) (Qi et al., 2020) for enhanced security during updates.
- **FedHGNN** (Yan et al., 2024): employs a federated heterogeneous graph neural network to explicitly capture the diverse structural relations between users and items for privacy-preserving recommendation.
- **FedSSP** (Tan et al., 2024): leverages spectral knowledge from client graphs to handle heterogeneity and models personalized preferences, adapted from federated graph classification.

## D.6 HYPERPARAMETER SETTINGS

We configure the hyperparameters of the proposed method as follows. All user, item, and popularity embeddings are initialized from a normal distribution, with an embedding dimension of 64. Optimization is performed using the RMSProp optimizer (Ruder, 2016) with a learning rate $\eta = 0.0005$.

Each client performs 5 local epochs of training per communication round. The entire training process consists of 40 global communication rounds, resulting in a total of 200 local training epochs per client. Each client performs 5 local epochs of training per communication round. The entire training process consists of 40 global communication rounds, resulting in a total of 200 local training epochs per client. The network architecture consists of two graph convolution layers, followed by two-layer MLPs used for both pooling and prediction stages (Eq. 4, 5). For the localized popularity bias-aware contrastive loss ($\mathcal{L}_{BC}$, Eq. 9), the softmax temperature parameter is set to $\tau = 0.1$, and the interpolation weight for the refined margin (Eq. 15) is set to $\omega = 0.25$. To ensure training stability, the model freezes parameter updates during the first two global epochs.

# E MODEL COMPLEXITY

## E.1 TIME COMPLEXITY ANALYSIS

### E.1.1 LOW-PASS GRAPH CONVOLUTION NETWORK'S TIME COMPLEXITY ANALYSIS

The propagation step in a Graph Convolution Network (GCN) can be expressed as $\mathbf{D}^{-\frac{1}{2}}\mathbf{A}\mathbf{D}^{-\frac{1}{2}}\mathbf{Z} = (\mathbf{I} - \mathbf{L})\mathbf{Z} = \mathbf{P}(\mathbf{I} - \mathbf{\Lambda})\mathbf{P}^\mathsf{T}\mathbf{Z}$, where it is equivalent to applying a low-pass filter in the frequency domain. The filter, represented as $[1 - \lambda_1, \ldots, 1 - \lambda_{M+N}]$, inherently prioritizes smaller eigenvalues, thereby emphasizing smooth, global features. Importantly, repeated applications of the Low-pass Collaborative Filter (LCF) are equivalent to a single application:

$$LCF(\ldots LCF(\mathbf{Z})) = (\bar{\mathbf{P}}\bar{\mathbf{P}}^\mathsf{T})^k \mathbf{Z} = (\bar{\mathbf{P}}\bar{\mathbf{P}}^\mathsf{T})\mathbf{Z} = LCF(\mathbf{Z}),$$

**Table 9:** Comparison of training efficiency across different models. Metrics include time per epoch, number of epochs to converge, total training time, and memory usage, with the best scores highlighted in **bold**.

| Model | Time/Epoch | #Epoch | Training Time | Memory Usage |
|---|---|---|---|---|
| FedAvg | 20s | **90** | **30m** | 6.2GB |
| FedPUB | 25s | **90** | 37m | 10.2GB |
| FedMF | 23m 48s | 150 | 59h 30m | 6.2GB |
| F2MF | 39m 12s | 100 | 65h 20m | 6.4GB |
| PFedRec | 38m 47s | 135 | 87h 16m | 1.4GB |
| FedRAP | 73m 29s | 160 | 195h 56m | 1.6GB |
| FedPerGNN | 6m 38s | 200 | 15h 29m | 18.8GB |
| FedHGNN | 36s | 100 | 1h | **0.6GB** |
| FedSSP | 20s | 150 | 50m | 6.2GB |
| LPSFed (BPR) | **19s** | 130 | 40m | 6.2GB |
| **LPSFed** | **19s** | 130 | 39m | 6.2GB |

ensuring stable feature propagation and avoiding over-smoothing.

Computing the full set of eigenvectors $\mathbf{P}$ for a graph with $M + N$ nodes has a time complexity of $\mathcal{O}((M + N)^3)$, as eigen-decomposition scales cubically. However, most real-world recommendation graphs are sparse, with the number of non-zero elements $n$ in the Laplacian $\mathbf{L}$ typically scaling linearly with $M + N$. By leveraging sparsity, the Lanczos algorithm computes a subset of eigenvectors $\bar{\mathbf{P}}$ for sparse matrices with time complexity $\mathcal{O}(n\Phi^2)$, where $\Phi$ represents the cut-off frequency (i.e., the number of retained low-frequency eigenvalues). In practice, where $\Phi \ll M + N$ and $n \ll (M+N)^2$, this approach ensures computational efficiency.

### E.1.2 LPSFed Framework's Complexity Analysis

In federated settings with $C$ clients, each client independently computes the first $\Phi$ eigenvectors using the Lanczos algorithm. The time complexity per client is $\mathcal{O}(n\Phi^2)$, where $n$ denotes the non-zero elements of the client's subgraph Laplacian. The server, which generates a global random graph, has a comparable computational complexity of $\mathcal{O}(n\Phi^2)$. Sequential execution would yield a total complexity of $\mathcal{O}((C+1)n\Phi^2)$. However, federated learning leverages parallel computation, allowing clients to perform computations simultaneously. As a result, the effective time complexity for the entire system remains $\mathcal{O}(n\Phi^2)$.

**Scalability and Robustness.** Our approach ensures scalability and robustness, addressing challenges in large subgraphs and imbalanced data distributions. The use of the Lanczos algorithm leverages the sparsity of real-world recommendation graphs, making eigenvector computations efficient even for large-scale datasets. Furthermore, pre-eigendecomposition ensures that training costs remain low, as this computational step is required only once.

**Communication and Computational Costs.** Communication costs in our framework depend on the size of the exchanged parameters, which occur only at specific aggregation and distribution epochs rather than every training iteration. This reduces the overall communication overhead. On the computational side, the server processes a graph representing the average subgraph size of clients, while each client processes its local subgraph, ensuring that both sides operate within their respective resource constraints. By limiting communication frequency and leveraging sparsity, our method balances computational and communication overheads, enabling high scalability and practical applicability.

### E.2 Training Efficiency Analysis

We compare the training efficiency and resource utilization (computation memory, time) of our model, LPSFed, with several other federated learning models. As shown in Table 9, we evaluate time per epoch, number of epochs to converge (#Epoch), total training time, and memory usage.

It's important to note that LPSFed, FedAvg, FedPUB, FedSSP are all based on the Low-pass Graph Convolution Network (LGCN) (Yu et al., 2022). This architecture allows for pre-computation

**Table 10:** Average Jaccard index for paired users over iterations, with rows corresponding to margin strengths $\gamma$ and columns to training iterations.

| Iteration | 20 | 40 | 60 | 80 | 100 |
|---|---|---|---|---|---|
| $\gamma = 0.0$ | 0.0482 | 0.0497 | 0.0536 | 0.0568 | 0.0600 |
| $\gamma = 0.3$ | 0.0478 | 0.0490 | 0.0529 | 0.0554 | 0.0592 |
| $\gamma = 0.5$ | 0.0477 | 0.0489 | 0.0522 | 0.0544 | 0.0576 |
| $\gamma = 0.7$ | 0.0476 | 0.0486 | 0.0512 | 0.0541 | 0.0570 |
| $\gamma = 1.0$ | 0.0474 | 0.0478 | 0.0501 | 0.0522 | 0.0553 |

of eigendecomposition, significantly reducing the computational burden during training. As a result, these models achieve faster training times compared to methods that require real-time graph convolutions. LPSFed demonstrates superior training efficiency, converging in 130 epochs with an average of 19 seconds per epoch, resulting in a total training time of 39 minutes and a memory usage of 6.2 GB. This shows an improvement over other models, emphasizing the efficiency of our approach. Leveraging the pre-computed convolution kernel allows our model to operate efficiently in the federated learning framework, capitalizing on the advantages of LGCN while optimizing performance through our proposed techniques, even in distributed settings.

# F   ADDITIONAL EXPERIMENTAL RESULTS

## F.1   BIAS AMPLIFICATION MEASUREMENT AND EFFECTIVENESS OF THE BIAS-AWARE MARGIN

**Bias Amplification Measurement.**    To assess how our method alleviates the feedback loop caused by popularity bias, we quantify convergence in user behavior using the Jaccard index, following (Chaney et al., 2018). This metric captures the extent to which recommender systems drive users to interact with increasingly similar items. For each user $u$, we identify the most similar user $v$ based on the cosine similarity of their preference vectors $(\theta_u, \theta_v)$ and compute the Jaccard index over their item interactions:

$$\mathbf{J}_{uv}(t) = \frac{|\mathcal{D}_u(t) \cap \mathcal{D}_v(t)|}{|\mathcal{D}_u(t) \cup \mathcal{D}_v(t)|},$$

where $\mathcal{D}_u(t)$ and $\mathcal{D}_v(t)$ denote the item sets interacted by users $u$ and $v$ up to time $t$. A higher Jaccard index reflects a stronger feedback loop, indicating behavioral convergence due to repeated exposure to popular items. To evaluate the impact of our localized popularity bias-aware margin, we vary the margin strength $\gamma$ and monitor changes in the Jaccard index over training. The results are presented in Table 10.

**Analysis.**    Table 10 demonstrates that the Jaccard index tends to increase over training iterations, indicating a typical feedback loop where users increasingly receive similar popular items. However, incorporating a bias-aware margin significantly mitigates this effect. As the margin strength $\gamma$ increases, the rate at which the Jaccard index grows is progressively reduced. Notably, when $\gamma = 1.0$, the index exhibits the smallest rise, suggesting that stronger margins more effectively suppress convergence driven by popularity.

These results suggest that the bias-aware margin provides a regularization signal that limits the reinforcement of frequently recommended items, thereby interrupting the progression of the feedback loop. By doing so, our method reduces recommendation redundancy and preserves both personalization and interaction diversity. This observation is consistent with prior findings in (Chaney et al., 2018), which report that unregulated feedback loops tend to amplify popularity-driven bias and degrade recommendation quality. In contrast, our approach effectively mitigates such dynamics, supporting more equitable and stable recommendation behavior across clients.

## F.2   CORRELATION ANALYSIS BETWEEN EIGENGAP AND PERFORMANCE

To examine whether spectral stability contributes to empirical performance, we computed the Pearson correlation between the observed Recall@20 values and the corresponding eigengaps $\delta$ at varying

**Table 11:** Correlation analysis between eigengap and model performance in *Movielens-1M*. Recall@20 and corresponding eigengap values are reported for different cut-off frequencies $\Phi$. Pearson correlation quantifies linear dependence between eigengap magnitude and model performance.

| Cut-off Frequency $\Phi$ | 2 | 4 | 6 | 8 |
|---|---|---|---|---|
| Recall@20 | 0.2556 | 0.2614 | 0.2615 | 0.2622 |
| Eigengap $\delta$ | $1.17 \times 10^{-15}$ | $4.36 \times 10^{-3}$ | $1.61 \times 10^{-2}$ | $1.46 \times 10^{-1}$ |
| Pearson Correlation (Recall $\leftrightarrow$ $\delta$) | **0.5004** | | | |

**Table 12:** Effect of client variability on performance using the *Amazon-Book* dataset. Metrics include Recall@20 and NDCG@20, **bold** indicates the best results and underlined the second-best results in each setting.

| # of Clients | 4 | | 10 | | 20 | |
|---|---|---|---|---|---|---|
| Model | Recall | NDCG | Recall | NDCG | Recall | NDCG |
| FedAvg | 0.0642 | 0.0312 | 0.0395 | 0.0186 | 0.0220 | 0.0098 |
| FedPUB | 0.0633 | 0.0322 | 0.0417 | 0.0180 | 0.0232 | 0.0102 |
| FedMF | 0.0153 | 0.0072 | 0.0091 | 0.0058 | 0.0033 | 0.0035 |
| F2MF | 0.0451 | 0.0225 | 0.0325 | 0.0182 | 0.0158 | 0.0078 |
| PFedRec | 0.0713 | 0.0242 | 0.0406 | 0.0169 | 0.0236 | 0.0100 |
| FedRAP | 0.0082 | 0.0090 | 0.0086 | 0.0085 | 0.0087 | 0.0074 |
| FedPerGNN | 0.0035 | 0.0026 | 0.0034 | 0.0026 | 0.0029 | 0.0025 |
| FedHGNN | 0.0647 | 0.0298 | 0.0415 | 0.0188 | 0.0223 | 0.0105 |
| FedSSP | 0.0649 | 0.0356 | 0.0450 | 0.0219 | 0.0238 | 0.0110 |
| LPSFed (BPR) | 0.0643 | 0.0322 | 0.0400 | 0.0195 | 0.0235 | 0.0107 |
| **LPSFed** | **0.0738** | **0.0442** | **0.0464** | **0.0259** | **0.0254** | **0.0115** |
| Improvement | ↑ 3.5% | ↑ 24.2% | ↑ 3.1% | ↑ 18.3% | ↑ 6.7% | ↑ 4.5% |

cut-off frequencies $\Phi$. As shown in Table 11, the correlation coefficient is 0.5004, indicating a moderate positive relationship. This trend is consistent with Figure 3 (c), where performance improves as $\Phi$ increases within a moderate range. The result supports our theoretical analysis; larger eigengaps $\delta$ enable more stable low-pass filtering by separating meaningful structural components from high-frequency noise, which contributes positively to model performance when applied within an appropriate filtering range.

### F.3 EFFECT OF CLIENT VARIABILITY ON PERFORMANCE

Table 12 presents the performance trends of all models under varying numbers of clients (4, 10, and 20) using the *Amazon-Book* dataset. As the number of clients increases, the overall performance of all baselines degrades. This degradation is attributed to the increasing heterogeneity among clients, which results from heightened *subgraph structural imbalance* and limited user-item interactions per client. These challenges amplify the difficulty of capturing consistent collaborative signals across clients, making recommendations more susceptible to data sparsity and structural diversity. Despite these challenges, LPSFed consistently maintains strong performance across all client settings. This robustness stems from two core components: spectral similarity-guided personalization and bias-aware margin. The former helps align model updates with each client's structural uniqueness, while the latter mitigates feedback loops caused by popularity bias. Together, they allow LPSFed to adapt to heterogeneous client environments and preserve recommendation quality, even as the number of clients grows and local data becomes more fragmented.

## G BROADER IMPACTS

This research on federated recommender systems provides several positive societal impacts. By training models on decentralized subgraph-level interaction data without exchanging raw data, it enhances privacy protection, reduces the risk of data exposure, and builds trust in domains where data sensitivity is critical, such as healthcare and finance. Our method also promotes fairness by mitigating localized popularity bias and improving the quality of recommendations for diverse client groups. Furthermore, the federated framework enables privacy-preserving collaboration across data silos, allowing clients to jointly improve recommendation quality without disclosing proprietary

data. These outcomes contribute to improved user experience, sustainable model utility, and the development of a more privacy-preserving recommendation ecosystem.

## H LIMITATIONS

Despite these promising advancements, our method inherently relies on spectral computations, such as eigendecomposition, performed during a preprocessing stage. Although preprocessing helps avoid computations during training, it still represents a scalability limitation. Future research should therefore prioritize developing scalable spectral approximation techniques and automating hyperparameter selection to further enhance the method's applicability in practical scenarios.

## I LLM USAGE

We utilized a Large Language Model (LLM) as an assistive tool in the preparation of this paper. The LLM's role was to proofread for grammar and spelling and to help refine the text for clarity, conciseness, and word choice. The core concepts, novel methodology, theoretical proofs, and all experimental results were conceived and generated by the human authors.

