# OpenReview forum: "Low-pass Personalized Subgraph Federated Recommendation"
_ICLR.cc/2026/Conference — ICLR 2026 Poster_

### Official Review · Reviewer_wXeM · 2025-10-25

**Soundness:** 3
**Presentation:** 3
**Contribution:** 3
**Rating:** 6
**Confidence:** 2

**Summary:**

In this paper, the authors introduce Low-Pass Personalized Subgraph Federated Recommendation (LPSFed) for personalized federated recommendation systems. LPSFed could address the challenge of subgraph structure imbalance and local popularity deviation. LPSFed uses low-pass spectral filtering on heterogeneous subgraphs to extract low-frequency structural signals. These signals are used to calculate the personalized structural similarity between each client and the global model. LPSFed also uses a local popularity bias-aware margin to capture the imbalance of item degrees in each subgraph and incorporates it into the personalized bias correction term to mitigate recommendation bias. Extensive results on multiple datasets demonstrate the effectiveness of the approach.

**Strengths:**

1. This paper is easy to follow and generally well presented.

2. It is a timely topic and a very interesting area to explore.

3. The theoretical analysis is solid.

**Weaknesses:**

1. The authors acknowledge in the Appendix and Limitations that spectral calculations are performed in the preprocessing stage. If the eigenvectors of the entire graph (or subgraph) are directly calculated on a very large graph (such as Tmall), it may not be feasible in terms of memory/time.

2. The author averages the local margins of all (u,i) of each client into a scalar $M^c$ and uploads it to the server. A single average value cannot reflect the skewness (such as long tail) or multimodal distribution within the client; the average value is sensitive to outliers and will mask important information.


3. Figure 1 shows a 15-client partition for Amazon-Book; however, D.2 in the main experiment states, "In the main experiment, we partitioned the full graph into four subgraphs." The paper does not clearly explain the correspondence between the different experimental settings and the reasons behind them.

**Questions:**

See in weaknesses.

---

> ### Author Response · Authors · 2025-11-21
>
> We thank the reviewer for your positive assessment of the clarity of our paper and the solidity of our theoretical analysis. We address your concerns regarding the scalability of spectral computation, information loss from margin averaging ($\mathcal M^c$), and the discrepancy in client partitioning (15 vs 4). We provided detailed replies to the weaknesses (W) below.
>
> > **W1. Spectral calculations in preprocessing may not be feasible on very large graphs in terms of memory/time (Also addresses Reviewer rJvg (W2, Q2)):**
>
> This is a valid concern and a common challenge for spectral methods. Our approach is designed specifically to mitigate this computational risk. Importantly, we never perform a full or dense factorization. Instead, as explained in Appendix E.1 (Model Complexity), we rely on two key strategies:
> 1. Efficient Preprocessing (Time and Memory): Our method avoids the prohibitive $\mathcal O((M+N)^3)$ time and $\mathcal O((M+N)^2)$ memory complexity of full eigendecomposition, where $M$ and $N$ are the number of users and items. Instead, as detailed in Appendix E.1.1, we leverage the sparsity of the graph and use the Lanczos algorithm.
> - Time: We compute only the first $\mathit\Phi$ eigenvectors ($\mathit\Phi \ll M+N$), reducing time complexity to $\mathcal O(n\mathit\Phi^2)$, where n is the number of non-zero elements.
> - Memory: We require only the $\mathcal O(n+(M+N)\mathit\Phi)$ space to store the sparse graph and the truncated eigenvectors. This is significantly more efficient than storing a dense matrix, making the processing step scalable even for large and sparse graphs.
>
> 2. One-Time Cost: Crucially, this $\mathcal O(n\mathit\Phi^2)$ computation is performed only once during preprocessing, before training begins. As shown in Table 9, completing this step in advance keeps the actual model training (Time/Epoch) efficient and comparable to standard FedAvg.
>
> However, for graphs at the scale of Tmall, we acknowledge that even the reduced $\mathcal O(n\mathit\Phi^2)$ preprocessing remains a substantial computational bottleneck. This is explicitly recognized as a key constraint in our Limitations (Appendix H).
>
> Looking ahead, we view this as an important opportunity for extension. The scalability of LPSFed can be further improved by adopting spectral approximation techniques, such as the Randomized SVD [1], which reduces complexity to $\mathcal O(n\mathit\Phi)$ and allows the parallelized matrix vector products. Integrating these methods would enable our framework to handle significantly larger industrial-scale graphs and further strengthen its practical applicability.
>
> **Writing Revision:** We revised Section 3.3 Low-pass Graph Filter \& Convolution (lines 153-158) to make these distinctions clearer using red text.
>
> [1] Nathan Halko et al., Finding structure with randomness: Probabilistic algorithms for constructing approximate matrix decompositions, SIAM Review, 2011.
>
> > **W2. Averaging local margins into a scalar value masks skewness and long-tail information:**
>
> The decision to average the margin into a single scalar $\mathcal M^c$ (Eq. 10) is intentional and reflects our priority to meet strict privacy constraints in Federated Recommender Systems (FRS).
>
> We agree that this aggregation loses detailed statistical characteristics. However, exposing more granular information, such as a full distribution or any $(u,i)$-level margin, would significantly increase privacy risks. Transmitting such data could enable the server to infer user interaction patterns or client-specific graph structures, which directly contradicts the privacy guarantees required in FRS.
>
> Therefore, $\mathcal M^c$ serves as the minimal yet privacy-safe summary of the client's bias tendency, used only to estimate the global bias context $\bar{\mathcal M}$ (Eq. 13). Despite its compressed form, empirical results in Table 3 (RQ3) demonstrate that this signal is sufficient; LPSFed consistently improves Tail, Mid, and Head performance beyond all baselines while maintaining privacy protection.
>
> **Writing Revision:** We revised Section 4.2 Training of Client Models (lines 255-263) to make these distinctions clearer using red text.

---

> ### Author Response · Authors · 2025-11-21
>
> > **W3. Figure 1 uses a 15-client partition, but the main experiment uses 4 clients; there is unclear correspondence and justification:**
>
> We apologize for the confusion regarding the client partition settings. The 15-client partition is used solely for motivational and illustrative purposes (Figure 1 and Table 2), whereas all main experiments and performance evaluations (Figure 3, Table 1, 3, 9, and 10) are conducted with a 4-client partition, following standard FRS benchmarking protocols. A clear mapping is provided below:
>
> |**Figure/Table**|**#Clients**|**Purpose**|
> |-|-|-|
> |Figure 1/Table 2|15|Motivational Analysis - To visualize the 'structural imbalance' phenomenon (**L**arge-**D**ense/**M**edium-**B**alanced/**S**mall-**S**parse groups) and demonstrate its impact on baseline performance.|
> |Figure 3/Table 1, 3, 9, 10|4|Main experiments - To conduct a fair and rigorous benchmark against baselines, perform ablations, and analyze sensitivity, following standard FRS protocols.|
>
> The 15-client setup provides a fine-grained visualization of heterogeneity (Figure 1), while the 4-client setup allows for a direct and fair comparison with prior FRS work (e.g., FedHGNN, FedPerGNN). We will update the captions for Figure 1 and Table 2 to clarify this distinction.
>
> **Writing Revision:** We revised the Figure 1 caption, Table 2 caption, and Section 5.1 Experimental Settings (lines 410-413) to make these distinctions clearer using red text.

---

> > ### Comment · Reviewer_wXeM · 2025-11-25
> >
> > Thanks for the authors' response, which addressed all my questions. I have no further concerns.

---

> > > ### Author Response · Authors · 2025-11-26
> > >
> > > Dear Reviewer wXeM,
> > >
> > > Thank you for your thoughtful review and constructive feedback. Your comments were very helpful in improving the clarity and rigor of the paper, and we have incorporated the suggested changes in the revised version.
> > >
> > > Thank you again for your valuable input.

---

### Official Review · Reviewer_7AW4 · 2025-10-31

**Soundness:** 3
**Presentation:** 2
**Contribution:** 3
**Rating:** 6
**Confidence:** 5

**Summary:**

The paper introduces LPSFed, a federated recommendation framework designed to handle structural imbalance across user–item subgraphs of different clients. This imbalance arises from variations in subgraph density and connectivity, which often lead to inconsistent representation learning when applying traditional GNN-based federated models. To address this issue, LPSFed employs low-pass spectral filtering to capture smooth, low-frequency graph signals, enhancing representation stability under heterogeneous structures. In addition, a popularity-aware contrastive loss is introduced to balance recommendations and reduce local popularity bias. Experiments on several datasets show consistent improvements over baseline federated models, and theoretical results provide further support for the effectiveness of the proposed spectral regularization strategy.

**Strengths:**

1. The paper offers a novel perspective by applying low-pass spectral filtering to mitigate structural imbalance in federated recommendation. The combination of spectral graph modeling and bias-aware optimization is original and novel.

2. The approach is conceptually clear. The appendix adds useful details on complexity analysis and spectral properties, improving transparency and reproducibility.

3. The study investigates an important but relatively underexplored problem in federated recommendation, namely the structural imbalance among decentralized user–item subgraphs. This imbalance is caused by variations in graph density and connectivity across clients. It can negatively influence the stability and fairness of model training in practical applications. By introducing spectral filtering and bias-aware optimization to address this issue, the paper provides a fresh perspective on improving structural consistency in federated recommender systems. The proposed direction may encourage further research on combining structure-aware learning with privacy-preserving mechanisms to develop more robust and equitable personalized recommendation models.

**Weaknesses:**

1. The core method of the paper is not sufficiently explained. The authors propose decomposing the embedding layer to generate a low-pass convolution kernel distribution k, which is then used for similarity comparison. However, the paper does not clarify why this specific k is chosen or why it is appropriate for measuring similarity. A more thorough discussion of both the theoretical and empirical reasoning behind this design would help readers understand its advantages over alternative approaches.

2. While the paper claims innovation in "structural imbalance," the uniqueness of the method compared to existing spectral federation methods (such as FedSSP) can be explained more clearly.

3. Although the description of the main method is relatively clear, the workflow diagram(Figure 2) appears somewhat cluttered, which makes it harder to fully understand the overall process.

**Questions:**

1.  Please provide a detailed explanation of the motivation behind the core method, specifically the similarity comparison based on the low-pass convolution kernel distribution k. The authors should clarify why this kernel distribution is chosen for measuring similarity and discuss its advantages over other possible approaches from both theoretical and empirical perspectives.

2. The model uses spectral domain filtering (Graph Fourier Transform + low-pass filtering) to remove high-frequency noise while preserving low-frequency structural information. Why is it better to retain the low-frequency signal for similarity comparison?

3. While the goal of federated learning is to preserve user privacy, the paper does not discuss whether the client-uploaded information (such as statistical features, spectral kernel distributions, or structural similarity scores) might pose a risk of indirectly revealing the underlying graph structure. A discussion or analysis of potential privacy implications would enhance the completeness of the work and align better with the core principles of federated learning.

---

> ### Author Response · Authors · 2025-11-21
>
> We thank the reviewer for highlighting the originality of combining spectral modeling with bias-aware optimization. Below, we address the concerns regarding the motivation for our kernel-based similarity, preference for low-frequency signals, potential privacy implications, comparison to FedSSP, and clarity of Figure 2. Detailed responses to each weakness (W) and question (Q) follow.
>
> > **W1, Q1, Q2 Motivation and choice of Low-pass kernel distribution $\mathbf K^c$:**
>
> The kernel distribution $\mathbf K^c$ (Eq. 6) is designed to provide a denoised, scale-invariant, and structurally meaningful representation of each client's subgraph. We adopt $D_{KL}(\mathbf K^R || \mathbf K^c)$ (Eq. 11) to compare structural signals reliably across clients. As raised in Q2, the design is based on the fundamental difference between low and high-frequency signals in graphs:
> 1. Low-frequency signals capture subgraph structure. They encode stable characteristics such as community organization and underlying topology [1], which reflect the essential structural semantics of subgraphs.
>
> 2. High-frequency signals largely correspond to noise. They reflect unstable, localized, and sparse interactions [2]. Comparing these signals across heterogeneous subgraphs (e.g., large-dense vs. small-sparse) introduces instability and exaggerates imbalance. Our low-pass filtering explicitly removes the high-frequency noise, enabling a stable comparison.
>
> 3. Normalization ensures scale invariance. Since raw eigenvalues $\mathbf\Lambda^c$ are strongly influenced by graph size and density, we normalize the filtered eigenvalues into a distribution $\tilde{\mathbf k}^c_{1:\mathit\Phi}$ (Eq. 6), eliminating their dependence on absolute magnitude and allowing fair comparison across clients.
>
> This design is supported by both theory and experiment:
> 1. Theoretical Justification (Theorem 4.1): We prove that $D_{KL}(\mathbf K^R || \mathbf K^c)$ reliably approximates the true structural divergence $D_{struct}$ and that this approximation is bounded by the eigengap $\delta$. This confirms that our metric captures intrinsic structural differences rather than noisy fluctuations.
> 2. Empirical Validation (Table 4 and Figure 3(c)): Table 4 (RQ4) shows a clear performance decline when omitting this component "w/o per", demonstrating its importance. Figure 3(c) further confirms Q2, performance drops when $\mathit\Phi$ is too large, indicating that including high-frequency signals weakens structural comparison.
>
> **Writing Revision:** We revised Section 3.3 Low-pass Graph Filter Convolution (lines 153-158), Section 4.1 Training of Client Models (lines 217-220), Section 4.4 Theoretical Analysis (lines 364-366), and Section 5.2 Experimental Results and Analysis (lines 521-525) to make distinctions clearer using red text.
>
> [1] U. Von Luxburg, A tutorial on spectral clustering, Statistics and Computing, 2007,
>
> [2] D. I. Shuman et al., The emerging field of signal processing on graphs, Signal Processing, 2013.

---

> ### Author Response · Authors · 2025-11-21
>
> > **W2. Novelty compared to existing spectral FL methods (Also addresses Reviewer SL1j W1; Reviewer rjvg W1, Q1):**
>
> While our approach builds on spectral principles, LPSFed is not an incremental adaptation of prior spectral FL methods such as FedSSP. The two works address fundamentally different problems and require distinct mechanisms.
>
> 1. Problem Setting: Structural Imbalance in FRS. FedSSP focuses on graph classification across homogeneous graphs with rich node features. In contrast, LPSFed addresses the distinct challenge of subgraph structural imbalance in FRS. We define this as the drastic disparity across client subgraphs in scale (user/item counts), connectivity density, and degree skewness, occurring within no node features, bipartite interaction graphs. FedSSP's mechanism, reliant on aligning node features, is simply not applicable to this FRS problem.
>
> 2. A New Personalization Mechanism based on Spectral Signals. Unlike FedSSP, which shares learned spectral knowledge through a global spectral encoder, LPSFed does not transmit any spectral or structural parameters across clients, thereby preventing privacy leakage. Instead, each client computes its own low-pass kernel distribution $\mathbf K^c$ (Eq. 6) to capture the intrinsic structural characteristics of its local subgraph, without revealing raw interactions or spectral parameters. The server generates a privacy-preserving random graph $G_R$ computes its spectral kernel $\mathbf K^R$ neutral structural anchor. We then measure the divergence via $D_{KL}$ (Eq. 11) to quantify structural imbalance, and use this as a personalized weight $\rho_c$ for aggregation.
>
> 3. Coupling with Bias Mitigation: Spectral Personalization and Bias Mitigation. Structural Imbalance in FRS is coupled with localized popularity bias. LPSFed is an integrated solution that synergistically couples our spectral personalization with a localized, privacy-preserving bias-aware margin $\mathcal M^c$ (Eq. 10).
>
> **Writing Revision:** We revised Section 1 Introduction (lines 85-92) and Section 2 Related Work (lines 104-107) to make these distinctions clearer using red text.
>
> > **W3, Q3. Figure 2 Clarity and Privacy Implications (Also addresses Reviewer SL1j (W3, Q1)):**
>
> We acknowledge that the original figure was dense, which limited the clarity of the information flow and the associated privacy safeguards. We will revise Figure 2 accordingly to emphasize this privacy-preserving protocol clearly.
>
> To address W3, we clarify that the proposed framework is designed to minimize privacy leakage by strictly limiting the information transmitted to the server. Importantly, the most sensitive spectral information (i.e., the full $\mathit\Phi$-dimensional kernel $\mathbf K^c$) remains strictly local and is never communicated. Instead, in each FL round, the client transmits:
>
> 1. Model Parameters $\theta^c$, consistent with standard FL methods.
> 2. Two aggregated, non-invertible scalars:
> - $\rho_c$ (Eq. 11): derived from the KL divergence, computed locally.
> - $\mathcal M^c$ (Eq. 10): the bias-aware margin, which is computed as the average of margin values over all nodes.
>
> Both $\rho_c$ and $\mathcal M^c$ are highly compressed one-way values and do not retain item- or node-level resolution. It is infeasible to reconstruct the client's graph $G_c$ or its spectral distribution from these scalars.
>
> Additionally, initial client statistics are shared only once to construct $G_R$, and are limited to high-level statistics (e.g., average node degree). Under stricter privacy constraints, this step can be omitted entirely. Our ablation ("w/o statistics for $G_R$", Table 4) directly evaluates this stricter setting and still achieves the second-best result, demonstrating that LPSFed remains effective even without any client statistics.
>
> **Writing Revision:** We revised Figure 2, Section 4.1 Training of Client Models (lines 255-263), and Section 4.2 Computing Structural Similarity (lines 270-274) to make distinctions clearer using red text.

---

### Official Review · Reviewer_rJvg · 2025-10-31

**Soundness:** 3
**Presentation:** 4
**Contribution:** 2
**Rating:** 4
**Confidence:** 3

**Summary:**

The paper targets the challenge of subgraph structural imbalance in FRS. The authors propose LPSFed which uses graph Fourier transform and low-pass spectral filtering to extract stable structural signals, measures structural similarity to guide personalization, and introduce a popularity-bias-aware margin to mitigate feedback loops. Experiments across five datasets show advantages supported by theoretical analysis.

**Strengths:**

The authors provide a well-motivated spectral view and lots of theoretical analysis. The qualitative tests show advantageous performance with discussions on robustness, completed by the ablation study of model components and hyperparameter analysis. The limitation of reliance on spectral computations is acknowledged.

**Weaknesses:**

1. Although the motivation of selected algorithms and theories are well motivated, the work seems to be a nice combination of existing proposals applied to this specific challenge, i.e., subgraph structural imbalance. This is not to undermine the effort and importance of such combination, but the novelty by itself is thus unavoidably limited.
2. As acknowledged by the authors in the appendix, the reliance on spectral computations raises the computation burden, I'd like to see if authors can provide some thoughts on how that could be alleviated.
3. It seems only one baseline is from the spectral FL family, please include more baselines from this direction, or otherwise explain the reason. The same applies to bias performance tests, which didn't compare with sufficient fairness-aware FRS works.
4. The cut-off frequency may have a considerable impact on system performance, yet the authors seem skip the discussion on the frequency value choice.  This also applies to the neutral structural anchor.
5. There seem to be quite some places in the math reasoning that are either vague or too idealized. For example, C1 and C2 only have vague description (actually only C1 has) without specifying how to determine their values; it idealizes the client graphs as having clear k-block community structures, which is quite rare in reality I'm afraid.

**Questions:**

Could the authors elaborate on the novelty besides combining existing algorithms and methods?
Could the authors provide some discussion on how the computation burden caused by spectral computation may be alleviated?
Can the authors include more fair comparisons with peering baselines such as spectral FRS and fairness-aware FRS?
Could the authors discuss more about the cut-off frequency and structural anchor determination and their impact and sensitivity?
Could the authors relax the idealization in math assumptions and give more specifics on value choices for example for C1 and C2 etc. ?

---

> ### Author Response · Authors · 2025-11-21
>
> We thank the reviewer for your constructive feedback and for recognizing our well-motivated spectral perspective, solid theoretical analysis, and strong performance. Below, we address your concerns on novelty, computational cost, additional spectral FL and fairness-aware baselines, hyperparameter sensitivity (cut-off frequency/anchor), and the assumptions behind our theoretical framework, with detailed replies to the weaknesses (W) and questions (Q).
>
> > **W1, Q1. Novelty compared to existing spectral FL methods (Also addresses Reviewer SL1j W1; Reviewer 7AW4 W2):**
>
> While our approach builds on spectral principles, LPSFed is not an incremental adaptation of prior spectral FL methods such as FedSSP. The two works address fundamentally different problems and require distinct mechanisms.
>
> 1. Problem Setting: Structural Imbalance in FRS. FedSSP focuses on graph classification across homogeneous graphs with rich node features. In contrast, LPSFed addresses the distinct challenge of subgraph structural imbalance in FRS. We define this as the drastic disparity across client subgraphs in scale (user/item counts), connectivity density, and degree skewness, occurring within no node features, bipartite interaction graphs. FedSSP's mechanism, reliant on aligning node features, is simply not applicable to this FRS problem.
>
> 2. A New Personalization Mechanism based on Spectral Signals. Unlike FedSSP, which shares learned spectral knowledge through a global spectral encoder, LPSFed does not transmit any spectral or structural parameters across clients, thereby preventing privacy leakage. Instead, each client computes its own low-pass kernel distribution $\mathbf K^c$ (Eq. 6) to capture the intrinsic structural characteristics of its local subgraph, without revealing raw interactions or spectral parameters. The server generates a privacy-preserving random graph $G_R$ computes its spectral kernel $\mathbf K^R$ neutral structural anchor. We then measure the divergence via $D_{KL}$ (Eq. 11) to quantify structural imbalance, and use this as a personalized weight $\rho_c$ for aggregation.
>
> 3. Coupling with Bias Mitigation: Spectral Personalization and Bias Mitigation. Structural Imbalance in FRS is coupled with localized popularity bias. LPSFed is an integrated solution that synergistically couples our spectral personalization with a localized, privacy-preserving bias-aware margin $\mathcal M^c$ (Eq. 10).
>
> **Writing Revision:** We revised Section 1 Introduction (lines 85-92) and Section 2 Related Work (lines 104-107) to make these distinctions clearer using red text.
>
> > **W2, Q2. Alleviating the Computation Burden (Also addresses Reviewer wXeM W1):**
>
> Our method alleviates the computational burden, a central concern for spectral approaches, by avoiding any full or dense decomposition. As detailed in Appendix E.1, it relies on two main principles:
> 1. Efficient Preprocessing (Time and Memory): Our method avoids the prohibitive $\mathcal O((M+N)^3)$ time and $\mathcal O((M+N)^2)$ memory complexity of full eigendecomposition, where $M$ and $N$ are the number of users and items. Instead, as detailed in Appendix E.1.1, we leverage the sparsity of the graph and use the Lanczos algorithm.
> - Time: We compute only the first $\mathit\Phi$ eigenvectors ($\mathit\Phi \ll M+N$), reducing time complexity to $\mathcal O(n\mathit\Phi^2)$, where n is the number of non-zero elements.
> - Memory: We require only the $\mathcal O(n+(M+N)\mathit\Phi)$ space to store the sparse graph and the truncated eigenvectors. This is significantly more efficient than storing a dense matrix, making the processing step scalable even for large and sparse graphs.
>
> 2. One-Time Cost: Crucially, this $\mathcal O(n\mathit\Phi^2)$ computation is performed only once during preprocessing, before training begins. As shown in Table 9, completing this step in advance keeps the actual model training (Time/Epoch) efficient and comparable to standard FedAvg.
>
> While this preprocessing may still pose a bottleneck for web-scale graphs (as noted in Limitations, Appendix H), the framework can be further scaled using randomized spectral approximation techniques such as Randomized SVD [1], which reduces complexity to $\mathcal O(n\mathit\Phi)$ and allows the parallelized matrix vector products. We consider an important direction for future work.
>
> **Writing Revision:** We revised Section 3.3 Low-pass Graph Filter \& Convolution (lines 153-158) to make these distinctions clearer using red text.
>
> [1] Nathan Halko et al., Finding structure with randomness: Probabilistic algorithms for constructing approximate matrix decompositions, SIAM Review, 2011.

---

> ### Author Response · Authors · 2025-11-21
>
> > **W3, Q3. Additional Spectral FL and Fairness-aware FRS baselines:**
>
> To address the request for broader comparison, we included additional baselines from both the fairness-aware FRS, F2PGNN [2], and the spectral FL, S2FGL [3], as representative methods. The results on Movielens-1M demonstrate that LPSFed consistently achieves the best performance among all compared models:
>
> |**Dataset**|**Movielens-1M**| |
> |-|-|-|
> |Model|Recall@20|NDCG@20|
> |FedAvg|0.2454|0.1240|
> |F2MF|0.1788|0.1120|
> |FedHGNN|0.2163|0.1031|
> |FedSSP|0.2564|0.1265|
> |F2PGNN (AAAI'24)|0.1830|0.1179|
> |S2FGL (ICML'25)|0.2522|0.1245|
> |**LPSFed**|**0.2646**|**0.1342**|
> |**Improvements**|**3.2$\%$**|**6.1$\%$**|
>
> These extended comparisons lead to two observations:
> 1. Fairness-aware FRS methods: Although these models, like F2MF and F2PGNN, explicitly aim to mitigate popularity bias, they do not incorporate structural personalization and therefore show reduced performance. Moreover, they rely on additional attributes (e.g., gender, age). By contrast, LPSFed captures bias through the aggregated scalar $\mathcal M^c$ without requiring any further information.
> 2. Spectral FL methods: While both FedSSP and S2FGL utilize spectral modeling, they apply shared spectral processing for all clients. LPSFed leverages spectral signals for personalized aggregation by introducing the structural alignment weight $\rho_c$ (Eq. 11), enabling client-specific adaptation. This leads to more consistent performance gain across heterogeneous client subgraphs.
>
> Overall, these results confirm that LPSFed not only inherits the strengths of spectral modeling but further improves upon them, enabling personalized adaptation while preserving privacy and maintaining robustness to popularity bias.
>
> [2] Nimesh Agrawal et al., No prejudice! Fair Federated Graph Neural Networks for Personalized Recommendation, AAAI, 2024,
>
> [3] Zihan Tan et al., S2FGL: Spatial Spectral Federated Graph Learning, ICML, 2025.
>
> > **W4, Q4. Analysis of Cut-off Frequency and Anchor Graph (Also addresses Reviewer SL1j (W4, Q2)):**
>
> We clarify the roles of both the cut-off frequency $\mathit\Phi$ and the neutral anchor graph, and summarize their effects using Figure 3(c) and Appendix D.3.
>
> 1. Cutoff Frequency $\mathit\Phi$ determines how many low-frequency eigencomponents are retained in the filtered kernel $\bar{\mathbf k}^c$ (Eq. 6). As shown in Figure 3(c), performance consistently peaks at $\mathit\Phi=128$:
>
> |**$\mathit\Phi$**|**8**|**16**|**32**|**64**|**128**|**256**|
> |-|-|-|-|-|-|-|
> |Imbalanced|0.0429|0.0430|0.0435|0.0438 |0.0442|0.0433|
> |Balanced|0.0357|0.0358|0.0373|0.0374|0.0390|0.0376|
>
> - Very small $\mathit\Phi$ fails to capture sufficient structural information.
> - $\mathit\Phi=128$ preserves the most informative low- and mid-frequency components while effectively filtering high-frequency noise.
> - Large $\mathit\Phi$ begins to introduce noisy high-frequency components, slightly degrading performance.
>
> This trend aligns with prior findings in spectral GNNs [4], where moderate low-pass filtering is most effective for sparse, degree-skewed bipartite graphs.
>
> 2. Sensitivity to Anchor Graph Design. First, the choice of anchor graph influences performance. The GNMK-based anchor consistently achieves better results than the ER model across all datasets, which aligns with our motivation that GNMK more accurately captures the structural properties of real-world recommendation graphs.
>
> Second, even though GNMK is stronger, LPSFed remains robust to different anchor configurations. The variant that excludes all client-derived statistics "w/o statistics for $G_R$", a setting meant to reflect stricter privacy constraints where no client-level statistics are available, constructs GNMK solely from global assumptions and still performs close to the full model. This demonstrates that our method does not rely heavily on detailed anchor information, even under more restrictive privacy settings.
>
> |**Dataset (Recall@20)**|**Amazon-Book**|**Gowalla**|**ML-1M**|**Yelp2018**|**Tmall-Buy**|
> |-|-|-|-|-|-|
> |ER|0.0734|0.1599|0.2631|0.0767|0.0410|
> |GNMK|0.0738|0.1621|0.2646|0.0783|0.0419|
> |w/o statistics for $G_R$ (GNMK)|0.0714|0.1581|0.2625|0.0764|0.0393|
>
> **Writing Revision:** We revised Section 5.2 Experimental Results and Analysis (lines 520-525) and added Appendix D.4 Sensitivity to Anchor Graph Design and Table 8 to make distinctions clearer using red text.
>
> [4] Felix Wu et al., Simplifying Graph Convolutional Networks, ICML, 2019.

---

> ### Author Response · Authors · 2025-11-21
>
> > **W5, Q5. Idealized Assumptions and Constants $C_1, C_2$:**
>
> 1. $C_1, C_2$ are Proof Constants, Not Algorithmic Hyperparameters.
> - $C_1$ in Lemma 2 arises from applying Weyl's inequality to the eigenvalues of each client subgraph. Let $\mathbf\Lambda^c=\text{diag}([\lambda^c_1,...,\lambda^c_{\mathit\Phi}])$ and $\bar{\mathbf k}^c=\mathbf\Lambda^c\odot\tilde{\mathbf f}_{1:\mathit\Phi}$ (Eq. 6). After normalization, the filtered spectrum becomes comparable across clients, and $C_1$ serves as a bound on how much this normalized spectrum may change under perturbations. Its definition is tied to the sum of non-zero eigenvalues in the filtered range and does not involve any tuning.
> - $C_2$ in Theorem 4.1 is the Lipschitz constant associated with the KL divergence between two normalized filtered eigenvalue distributions. Because low-pass filtering keeps only the first $\mathit\Phi \ll M+N$ eigenvalues and normalization ensures strictly positive elements, the constant remains finite and well-defined. Neither $C_1$ nor $C_2$ is selected or tuned during training; both follow directly from standard perturbation and information-theoretic arguments applied to the filtered Laplacian spectrum.
>
> 2. Idealized Assumptions and Latent Structure. Our theory (Assumption 1) models each client subgraph as having an underlying community structure, but not necessarily a perfect $k$-block model. Theorem 4.2 uses the eigengap $\delta=\lambda_{\mathit\Phi}-\lambda_{\mathit\Phi+1}$ to bound representation variance after low-pass filtering, showing that filtering suppresses high-frequency noise even in irregular real-world bipartite graphs. The $k$-block assumption is therefore a sufficient condition for analysis, not a requirement for the method to work in practice.
>
> 3. Empirical Support and No Manual Tuning. Table 2 and Figure 1 (c) show that the method performs consistently across all subgraph types (Large-Dense, Medium-Balanced, Small-Sparse). All spectral quantities, such as normalized eigenvalues, eigengaps, and filtered kernels, are computed automatically during preprocessing without any manual adjustment. The algorithm adapts to graph heterogeneity by construction.
>
> **Writing Revision:** We revised Section 4.4 Theoretical Analysis (lines 364-366) to make distinctions clearer using red text.

---

### Official Review · Reviewer_SL1j · 2025-11-02

**Soundness:** 3
**Presentation:** 3
**Contribution:** 3
**Rating:** 6
**Confidence:** 3

**Summary:**

This paper introduces LPSFed, a federated recommender framework designed to handle subgraph structural imbalance and popularity bias across clients. It applies low-pass spectral filtering to extract stable, low-frequency structural signals and computes a structural similarity between each client and a server-generated reference graph for personalized aggregation. Additionally, it introduces a bias-aware margin loss that mitigates popularity bias by adjusting item ranking based on local degree distributions. Theoretical analysis justifies the stability and regularization of low-pass filtering, and experiments on five real-world datasets demonstrate consistent improvements in recall, NDCG, and robustness over multiple baselines.

**Strengths:**

- Clear and coherent framework integrating personalization and debiasing.

- Sound theoretical motivation for spectral similarity and low-pass regularization.

- Strong empirical results across multiple datasets and settings.

- Effective ablation studies confirming each module’s contribution.

**Weaknesses:**

- Novelty is incremental; relies heavily on previous spectral FL ideas.

- Theoretical assumptions may not align with real-world bipartite graphs.

- Ambiguity in privacy handling (what information the server receives).

- Sensitivity to anchor graph design and data partitioning not deeply analyzed.

**Questions:**

1 Can similarity computation be done entirely on clients to ensure privacy?

2 How sensitive is performance to the reference graph’s design or randomness?

3 Do empirical eigengaps correlate with observed improvements?

4 Can the bias-aware margin adapt automatically rather than be fixed?

5 Has the model been tested under naturally partitioned (non-synthetic) clients?

---

> ### Author Response · Authors · 2025-11-21
>
> We thank the reviewer for recognizing the clarity of our framework, the strength of theoretical motivation, and the empirical performance of LPSFed. We address the key concerns regarding novelty, theoretical assumptions, privacy considerations, and reference graph design. Below, we provide detailed responses to each weakness (W) and question (Q) raised.
>
> > **W1. Novelty is incremental; it relies heavily on previous spectral FL ideas (Also addresses Reviewer rJvg W1, Q1; Reviewer 7AW4 W2):**
>
> While our approach builds on spectral principles, LPSFed is not an incremental adaptation of prior spectral FL methods such as FedSSP. The two works address fundamentally different problems and require distinct mechanisms.
>
> 1. Problem Setting: Structural Imbalance in FRS. FedSSP focuses on graph classification across homogeneous graphs with rich node features. In contrast, LPSFed addresses the distinct challenge of subgraph structural imbalance in FRS. We define this as the drastic disparity across client subgraphs in scale (user/item counts), connectivity density, and degree skewness, occurring within no node features, bipartite interaction graphs. FedSSP's mechanism, reliant on aligning node features, is simply not applicable to this FRS problem.
>
> 2. A New Personalization Mechanism based on Spectral Signals. Unlike FedSSP, which shares learned spectral knowledge through a global spectral encoder, LPSFed does not transmit any spectral or structural parameters across clients, thereby preventing privacy leakage. Instead, each client computes its own low-pass kernel distribution $\mathbf K^c$ (Eq. 6) to capture the intrinsic structural characteristics of its local subgraph, without revealing raw interactions or spectral parameters. The server generates a privacy-preserving random graph $G_R$ computes its spectral kernel $\mathbf K^R$ neutral structural anchor. We then measure the divergence via $D_{KL}$ (Eq. 11) to quantify structural imbalance, and use this as a personalized weight $\rho_c$ for aggregation.
>
> 3. Coupling with Bias Mitigation: Spectral Personalization and Bias Mitigation. Structural Imbalance in FRS is coupled with localized popularity bias. LPSFed is an integrated solution that synergistically couples our spectral personalization with a localized, privacy-preserving bias-aware margin $\mathcal M^c$ (Eq. 10).
>
> **Writing Revision:** We revised Section 1 Introduction (lines 85-92) and Section 2 Related Work (lines 104-107) to make these distinctions clearer using red text.
>
> > **W2. Theoretical assumptions may not align with real-world bipartite graphs:**
>
> We agree that our theoretical analysis (Section 4.4 Assumption 1) relies on an idealized $k$-community structure for mathematical tractability. The intention of this assumption is not to replicate the full complexity of real-world bipartite graphs, but to formally justify the key components of our method.
>
> - Theorem 4.1 demonstrates that the KL-based kernel divergence provides a well-behaved and bounded measure of structural discrepancy, supporting its use for personalization.
> - Theorem 4.2 shows that low-pass filtering acts as a stable spectral regularizer, making representations resistant to structural noise and sparsity differences across clients.
>
> Although real user-item interaction graphs are not perfect $k$-block structures [1], they do exhibit meaningful latent communities and low-frequency patterns. Our extensive experiments on five real-world FRS datasets (Table 1) consistently show that LPSFed outperforms all baselines, indicating that the theoretical properties we analyze indeed translate to practical benefits.
>
> **Writing Revision:** We revised Section 4.4, Theoretical Analysis (lines 364-366) to make these distinctions clearer using red text.
>
> [1] Maksim Kitsak et al., Latent geometry of bipartite networks, Physical Review E, 2017.

---

> ### Author Response · Authors · 2025-11-21
>
> > **W3, Q1. Ambiguity in privacy handling (Also addresses Reviewer 7AW4 (W3, Q3)):**
>
> LPSFed is designed to minimize privacy leakage by strictly limiting the information transmitted to the server. Below, we clarify the exact communication protocol.
>
> 1. Per-round Communication: Only Minimal, Non-invertible Information. In each FL round, the client sends only:
> - Model parameters $\theta^c$: the standard weight updates used in all FL frameworks.
> - Two Non-Invertible Scalars: The personalization weight $\rho_c$ (Eq. 11), computed from the local KL divergence. The bias-aware margin summary $\mathcal M^c$ (Eq. 10) is an averaged statistic of local bias terms.
>
> Crucially, the server never receives the $\mathit\Phi$-dimensional spectral kernel $\mathbf K^c$ nor any item- or node-level information. Since both $\rho_c$ and $\mathcal M^c$ are aggregated scalars, it is not feasible to reconstruct the client's graph $G_c$ or its spectral distribution from these values.
>
> 2. Clarification on Optional Initialization Statistics. We acknowledge that the mention of high-level client statistics (e.g., average degrees) may have caused confusion. These statistics are part of an optional one-time initialization used solely to inform the construction of the neutral anchor $G_R$. This setting removes all client-provided statistics, enforcing a stricter privacy regime. Despite this constraint, the model maintains robust performance. This confirms that:
> - LPSFed is less dependent on client-side structural information.
> - The method remains effective even under high-privacy constraints.
>
> **Writing Revision:** We revised Figure 2, Section 4.1 Training of Client Models (lines 255-263), and Section 4.2 Computing Structural Similarity (lines 270-275) to make distinctions clearer using red text.
>
> > **W4, Q2, Q5. Analysis of Anchor Graph and Data Partitioning (Also addresses Reviewer rJvg (W4, Q4)):**
>
> 1. Sensitivity to Anchor Graph Design (W4, Q2). Our analysis in Appendix D.3 reveals two key findings. First, the design of the reference graph does affect performance. Across all datasets, the GNMK-based anchor consistently outperforms the ER model, supporting our motivation that GNMK better captures the structural properties of real-world recommendation graphs.
> Second, despite this difference, LPSFed remains robust to variations in the anchor configuration. The "w/o statistics for $G_R$" variant, which constructs the GNMK anchor without using any client-side statistics, achieves performance close to the full model. This confirms that the method does not depend on detailed client information and maintains stability even under stricter privacy settings. These results show that GNMK is the better choice, while LPSFed as a whole remains stable under different anchor designs.
>
> |**Dataset (Recall@20)**|**Amazon-Book**|**Gowalla**|**ML-1M**|**Yelp2018**|**Tmall-Buy**|
> |-|-|-|-|-|-|
> |ER|0.0734|0.1599|0.2631|0.0767|0.0410|
> |GNMK|0.0738|0.1621|0.2646|0.0783|0.0419|
> |w/o statistics for $G_R$ (GNMK)|0.0714|0.1581|0.2625|0.0764|0.0393|
>
> **Writing Revision:** We added Appendix D.4 Sensitivity to Anchor Graph Design and Table 8 to make distinctions clearer using red text.
>
> 2. Analysis of Data Partitioning (W4, Q5). Publicly available, naturally partitioned FRS datasets are limited. For this reason, our original benchmark was constructed using spectral clustering, not as a synthetic random split, but as a principled approach that preserves community structure and increases degree variance (Noted in Appendix D.2 and Table 7). This approach generates structurally heterogeneous subgraphs that effectively reflect the realistic data imbalances targeted by LPSFed. We agree, however, that evaluation on a fully natural partition is ideal. Following the reviewer's suggestion, we conducted an additional experiment using Yelp2022 [2], which includes region metadata. We selected four major regions (PA, FL, LA, TN) to create naturally distinct client subgraphs that reflect real-world structural differences. As shown below, LPSFed again achieves the best overall performance and surpasses all baselines, including strong spectral methods such as FedSSP. This confirms that LPSFed remains robust and effective under genuine, naturally partitioned FRS conditions.
>
> |**Dataset**|**Yelp2022**| |
> |-|-|-|
> |Model|Recall|NDCG|
> |FedAvg|0.0813|0.0373|
> |FedPUB|0.0822|0.0385|
> |FedMF|0.0368|0.0180|
> |FedPerGNN|0.0329|0.0184|
> |FedHGNN|0.0815|0.0387|
> |FedSSP|0.0834|0.0428|
> |LPSFed (BPR)|0.0833|0.0412|
> |**LPSFed**|**0.0860**|**0.0467**|
>
> [2] Yelp2022 dataset, https://www.kaggle.com/datasets/yelp-dataset/yelp-dataset, Accessed: 2025-11-20.

---

> ### Author Response · Authors · 2025-11-21
>
> > **Q3. Correlation between eigengaps and model performance:**
>
> The following table presents the Movielens-1M results for different cut-off frequencies $\mathit\Phi$, showing both the Recall@20 and the corresponding eigengap values $\delta$.
>
> |**Cut-off Frequency $\mathit\Phi$**|**2**|**4**|**6**|**8**|
> |-|-|-|-|-|
> |Movielens-1M (Recall@20)|0.2556|0.2614|0.2615|0.2622|
> |Eigengap $\delta$|1.17e-15|4.36e-3|1.61e-2|1.46e-1|
> |Pearson Correlation Recall $\leftrightarrow \delta$| | 0.5004 | | |
>
> We calculated the Pearson correlation coefficient between Recall and the corresponding eigengap values $\delta$ to examine their relationship. Pearson correlation quantifies the strength of linear dependence between two variables (-1 to +1). The result of 0.5004 indicates a moderate positive correlation.
>
> This observation aligns with Figure 3(c), as $\mathit\Phi$ increases within a reasonable range, performance improves. From a spectral perspective [3, 4], a larger eigengap indicates clearer structural separation, enabling low-pass filtering to preserve informative global patterns while effectively suppressing high-frequency noise. However, when $\mathit\Phi$ grows too large (e.g., beyond 128), high-frequency components begin to dominate, diminishing the overall gain.
>
> Overall, the results support our theoretical analysis that spectral stability, reflected by larger eigengaps $\delta$, contributes to improved generalization performance, but only when filtering remains within an appropriate range. This correlation analysis has been added to Appendix F.2 and Table 10.
>
> [3] Ron Levie et al., Transferability of Spectral Graph Convolutional Neural Networks, JMLR, 2021,
>
> [4] Hoang-Son Nguyen, On the Stability of Low-pass Graph Filter with a Large Number of Edge Rewires, ICASSP, 2022.
>
> > **Q4. Adaptive Bias-aware Margin:**
>
> We evaluated an auto-margin variant where the margin scaling factor $\nu$ is learned rather than fixed. The learnable parameter $\nu$ is initialized to 1.0, constrained to remain positive. The results on the Amazon-Book dataset are as follows:
>
> |**Dataset**|**Amazon-Book**| |
> |-|-|-|
> |Model|Recall|NDCG|
> |LPSFed(BPR)|0.0643|0.0322|
> |LPSFed(Auto Margin)|0.0741|0.0442|
> |LPSFed|0.0738|0.0442|
>
> - Empirical Effect: Learning the margin provided a slight gain in Recall, while the NDCG score remained unchanged. The slight increase in recall appears to result from optimization dynamics, although the overall ranking quality remains unaffected.
>
> - Stability and Cost: The learnable-margin variant showed higher sensitivity during optimization and required additional tuning of the learning rate and temperature to ensure stable convergence. This increases the risk of instability when applied under realistic federated settings where privacy constraints and non-IID data distributions already challenge learning consistency.
>
> In summary, while learnable margins show potential for slightly improving Recall, they introduce additional parameter tuning and may negatively affect training stability. We therefore adopt a fixed approach in this work, as it provides consistent bias control and stable performance across all settings, which is supported by the results in Figure 3 and Table 2.

---

### Author Response · Authors · 2025-12-02
**Thank You and Summary of Revisions for Paper #15916**

Dear Area Chair,

We appreciate your efforts in managing the review process and the reviewers for their constructive feedback on our paper, **Low-pass Personalized Subgraph Federated Recommendation (LPSFed)**. The reviewers highlighted several strengths of LPSFed, including the soundness of our theoretical analysis and the strong empirical performance (Reviewers SL1j, rJvg, 7AW4, and wXeM). Notably, 7AW4 highlighted the novelty of addressing structural imbalance as an underexplored challenge in FRS.

To address the concerns raised, we have conducted six additional experiments and revised the paper. Below, we summarize the major improvements, detailing each concern (C) and our corresponding revision (R).

> # 1. Clarifying Novelty and Differentiation (vs. Spectral FL)
- C1: The method may be incremental compared to existing spectral FL (Reviewers SL1j, rJvg, and 7AW4).
- R1:
  - We clarified that prior spectral FL (e.g., FedSSP) targets graph classification and node feature alignment, whereas LPSFed deals with subgraph structural imbalance in FRS, characterized by featureless bipartite graphs with highly scale-density disparities.
  - Unlike spectral feature alignment, our method models the low-pass kernel distribution $\mathbf K^c$ (where $c$ denotes the client) as a scale-invariant structural signal and derives personalized weights using a new spectral similarity metric $D_{KL}$ (Revised Sec. 1 and 2).

> # 2. Connecting Theory with Real-World Scenarios
- C2: Idealized assumptions (e.g, $k$-block communities) may not fully reflect real-world graph structures (Reviewer SL1j and rJvg).
- R2: We clarified that these assumptions are not intended to model real-world graphs perfectly, but rather to provide formal justification for the stability of our design choices. In particular, the theoretical analysis demonstrates that low-pass filtering is a spectral regularizer and $\rho_c$ is a stable similarity metric even under structural noise. While real-world graphs may deviate from idealized conditions, LPSFed consistently outperforms all baselines across five large-scale datasets, confirming that the proposed mechanism remains effective in practical settings (Revised Sec. 4.4).

> # 3. Improving Clarity on Privacy and Information Flow
- C3: Ambiguity in parameter sharing and potential privacy risks (Reviewers SL1j, 7AW4, and wXeM).
- R3:
  - We clarified that the spectral kernel $\mathbf K^c$ never leaves the client. During each communication round, the client only sends (1) the local model parameters $\theta_c$ and (2) two non-invertible scalars $\rho_c$ and $\mathcal M^c$.
  - As further clarified in the revision, initial local subgraph statistics used to configure the anchor graph can be optionally exchanged once before training. However, this step is not required for our method to function. We validated this via the "w/o statistics for $G_R$" ablation, which removes all such statistics for stricter privacy and still delivers competitive results. (Revised Sec. 4.1 and 4.2)
  - Figure 2 was redesigned to visualize this privacy-preserving workflow better.

> # 4. Enhanced Experimental Validation
- C4: The use of spectral clustering for client partitioning, the sensitivity to anchor graph design, the range of baselines (including spectral FL and fairness-aware FRS), and the empirical link between eigengap $\delta$ and performance (Reviewers SL1j, rJvg, and 7AW4).

- R4:
  - Natural Partitioning Experiment: We added a new experiment using the Yelp2022 dataset partitioned by real-world regions (PA, FL, LA, TN), demonstrating that LPSFed remains robust and effective in naturally partitioned environments.
  - Anchor Design Sensitivity: We analyzed the impact of anchor graph design by comparing GNMK, GNMK without using any client statistics, and ER models. Results confirm that the framework is robust to anchor variations.
  - Additional Baselines: We incorporated comparisons against S2FGL and F2PGNN, showing that LPSFed consistently outperforms state-of-the-art methods from these related domains.
  - Eigengap Correlation Analysis: We verified our theoretical stability bounds by demonstrating a positive correlation (Spearman) between the empirical eigengap and model performance.

> # 5. Scalability and Computational Efficiency
- C5: Computational burden of spectral operations on large graphs (Reviewers rJvg and wXeM).
- R5:
  - We emphasized that our use of the Lanczos algorithm on sparse graphs reduces preprocessing complexity to $\mathcal O(n\mathit\Phi^2)$ (where $\mathit\Phi$ denotes the cut-off frequencies and $n$ is the number of non-zero elements), which is a one-time cost before training.
  - We discussed future scalability improvements, such as Randomized SVD.

Through this revision, we have significantly strengthened the theoretical justification, empirical rigor, and clarity of our work. We appreciate the reviewers and the AC for your valuable time and insights.

Best regards, The Authors.

---

### Meta-Review · Area_Chair_MBrh · 2026-01-07

**Summary:**

1. The method may be incremental compared to existing spectral FL methods.
2. Idealized assumptions may not fully reflect real-world graph structures.
3. Computational burden of spectral operations on large graphs.
4. More details of experiments are needed, such as spectral clustering, parameter sensitivity, and more baselines.

**Reviewer Concerns:**

All points were partially addressed.

**Reviewer Scores:**

No change

---

### Decision · Program_Chairs · 2026-01-26

Accept (Poster)